# Synthetic accessibility and stability rules of NASICONs

Bin Ouyang[1,2,4], Jingyang Wang[1,2,4], Tanjin He[1,2], Christopher J. Bartel [1,2], Haoyan Huo [1,2], Yan Wang [3], Valentina Lacivita[3], Haegyeom Kim[1] & Gerbrand Ceder [1,2✉]

In this paper we develop the stability rules for NASICON-structured materials, as an example of compounds with complex bond topology and composition. By first-principles high-throughput computation of 3881 potential NASICON phases, we have developed guiding stability rules of NASICON and validated the ab initio predictive capability through the synthesis of six attempted materials, five of which were successful. A simple two-dimensional descriptor for predicting NASICON stability was extracted with sure independence screening and machine learned ranking, which classifies NASICON phases in terms of their synthetic accessibility. This machine-learned tolerance factor is based on the Na content, elemental radii and electronegativities, and the Madelung energy and can offer reasonable accuracy for separating stable and unstable NASICONs. This work will not only provide tools to understand the synthetic accessibility of NASICON-type materials, but also demonstrates an efficient paradigm for discovering new materials with complicated composition and atomic structure.

[1] Materials Sciences Division, Lawrence Berkeley National Laboratory, Berkeley, CA, USA. [2] Department of Materials Science and Engineering, University of California, Berkeley, CA, USA. [3] Advanced Material Lab, Samsung Research America, 10 Wilson Rd., Cambridge, MA, USA. [4] These authors contributed equally: Bin Ouyang, Jingyang Wang. ✉email: gceder@berkeley.edu

The advancement of high-throughput computation and data-mining techniques[1,2] enables us to rapidly search for new materials with interesting properties[1,3–6], yet its success has now become limited by our lack of knowledge of which predicted new materials are experimentally accessible. A trial-and-error synthesis approach is tedious, if not intractable, due to the exponential scaling of compositional possibilities in multi-component spaces[7]. Therefore, the development of a rapid approach to evaluate the stability of compounds with complicated composition and structure would be a crucial step to accelerate materials discovery.

The stability of a target compound is a nonlocal property in the compositional space of interest, as it involves competitions between the energy of the target, and many possible combinations of competing phases, often at different compositions[8,9]. For this reason, models that solely focus on reproducing the formation energy of a compound have limited ability to predict its stability[10], while phenomenological models which take physical factors into account seem to be a better option. Synthesizability is hard to predict because it is a fundamentally nonlocal property[10], but it can nonetheless be learned using only local descriptors, specifically when the scope of materials under investigation is restricted to a single structure or class of structures[11]. For example, one of the best-known models, the Goldschmidt tolerance factor proposed by Victor Goldschmidt in 1926[12], uses the ratio of ionic radii to estimate the stability of a perovskite. Similar models have been formulated for other materials with relatively simple composition and bonding topology[11,13–18]. However, for multicomponent compounds with a complex crystal structure, it is challenging to predict the stability with a simple physical factor such as ionic radius, electronegativity, or by empirical rules, such as Pauling's rules[19].

Sodium (Na) SuperIonic Conductor (NASICON) compounds are a good example of materials with complex composition and bond topology. NASICONs have the general chemical formula $Na_xM_2(AO_4)_3$ and consist of a 3D corner-sharing framework made of $AO_4$ tetrahedra and $MO_6$ octahedra (Fig. 1a). Within the polyanion framework, there are four different sites for cations, as illustrated in Fig. 1b, while all oxygens sit in the 36 f Wyckoff positions. Na content ($x$) can vary from 0 to 4, and a variety of cations can coexist in both the M site and A site (Fig. 1c)[20–32]. Given such a compositional and bonding complexity, multiple physical factors, including propensity for an element to take a particular coordination environment, and the bond compatibility among different site environments, can affect the phase stability. As a result, the stability rules for NASICON materials are likely to be nontrivial. NASICON materials are also of significant technological interest as they exhibit high ionic conductivity, which is favorable in many applications, including selective ion membranes, gas sensing devices, and Na-ion batteries (as both electrodes and solid-state electrolytes)[33–39]. Since the first report of the prototype NASICON $Na_3Zr_2(SiO_4)_2(PO_4)$ in the 1960s[29], most of the successfully synthesized new materials are compositionally similar to this original, as illustrated in Fig. 1c, and a broad search for other NASICONs could be challenging due to the large possible compositional space.

In the current work, we developed and applied a suite of materials discovery, physical interpretation, and machine-learning tools to uncover the stability rules of NASICON-type materials across a broad compositional space. We first performed high-throughput phase diagram calculations to sample the energy of NASICON compounds in the chemical space Na-M$_1$-M$_2$-A-B-O. The high-throughput screening covered 21 metal elements and evaluated 3881 compositions, only 32 of which have been previously explored. Guided by the high-throughput phase stability predictions, we successfully synthesized five out of six predicted new NASICONs in

the Na-rich silicon phosphate subspace. We also rationalized the stability trend of NASICON in terms of bond compatibility and site miscibility. Finally, to enable the prediction of stability from basic physical properties, we combined Sure Independence Screening (SIS)[40–42] and machine-learned ranking (MLR)[43,44] to identify the best 2D descriptors that can separate NASICONs that are likely to be synthesized from unstable NASICONs. With the constructed features, $t1 = \sqrt[3]{N_{Na}} + (Q_A^{Std})^2$ and $t2 = E_{Ewald}^2 \bullet X_M^{Na} \bullet R_M^{Std}$, where $N_{Na}$ is Na content, $X_A^{Std}$ the standard deviation of A site electronegativity, $E_{Ewald}^2$ the Ewald summation of the electrostatic energy, $X_M^{Na}$ the electronegativity difference between M site and Na, and $R_M^{Std}$ the standard deviation of the radius in M site, the synthetic accessibility with solid-state synthesis at 1000 K can be estimated using the simple relationship: $0.203 \times t1 + t2 \le 0.322$.

The tolerance factors we developed for NASICONs is one of the first machine-learned phenomenological models to estimate the stability of complex oxides within a large compositional space. Our work will thus not only facilitate the search for NASICON-type materials but also from a broader perspective, provide insights to help apply state-of-the-art machine-learning techniques and data-mining tools to accelerate the discovery of new inorganic materials with complicated compositions and bond topologies.

## Results

**Stability map of NASICON based on high-throughput DFT calculations.** To computationally explore NASICONs with the chemical formula $Na_xM_yM'_{2-y}(AO_4)_z(BO_4)_{3-z}$, we sampled two compositional variables: the Na content $x$ from 0.0 to 4.0 (in steps of 0.5) and the average M/M' charge state from +2 to +5 (in steps of 0.5). For each sampled Na content and M/M' charge state, all possible combinations of $(SiO_4)^{4-}$, $(PO_4)^{3-}$, and $(SO_4)^{2-}$ polyanions that result in a charge-balanced composition were enumerated, leading to 3881 possible NASICON compounds. For each NASICON composition, the Ewald energy of different configurational arrangements of the Na ions, metal ions, and polyanions was determined and the one with the lowest electrostatic energy was computed with DFT. The relative stability of each NASICON compound is quantified by the energy above the convex hull ($E_{hull}$)[1]. The $E_{hull}$ value of a novel NASICON is calculated with respect to the convex hull formed by all the compounds in the Materials Project in the relevant chemical space (not including the NASICON). A negative $E_{hull}$ value therefore indicates that the computed NASICON is a new ground state.

In Fig. 2, the effect of chemical composition on the phase stability of the NASICON compounds is characterized with two factors: the median of the $E_{hull}$ values in the relevant compositional space and the frequency ($f_{accessible}$) by which the $E_{hull}$ of the NASICON compound is less than $S_{ideal} \times 1000\ K$, where $S_{ideal}$ represents the ideal entropy of mixing as explained in supplementary note 1. The second factor gauges which NASICONs are close enough to the 0 K hull so that they may be stabilized by configurational entropy at the synthesis temperature. It should be noted that, though not equivalent to synthetic accessibility, $E_{hull}$ has been demonstrated to be a reasonable estimate of synthetic accessibility[45].

In Fig. 2a, the lower triangle shows the median $E_{hull}$ values for all the NASICONs that contain both the metal on the $x$-axis and the metal on the $y$-axis, while the upper triangle shows the corresponding $f_{accessible}$ values. Figure 2b maps the median $E_{hull}$ values and $f_{accessible}$ for all the NASICONs as a function of Na content and polyanion composition. The results in Fig. 2a indicate that NASICONs containing metals such as $Hf^{4+}$, $Zr^{4+}$, $Ta^{5+}$, and $Sc^{3+}$ have higher stability and are more compatible

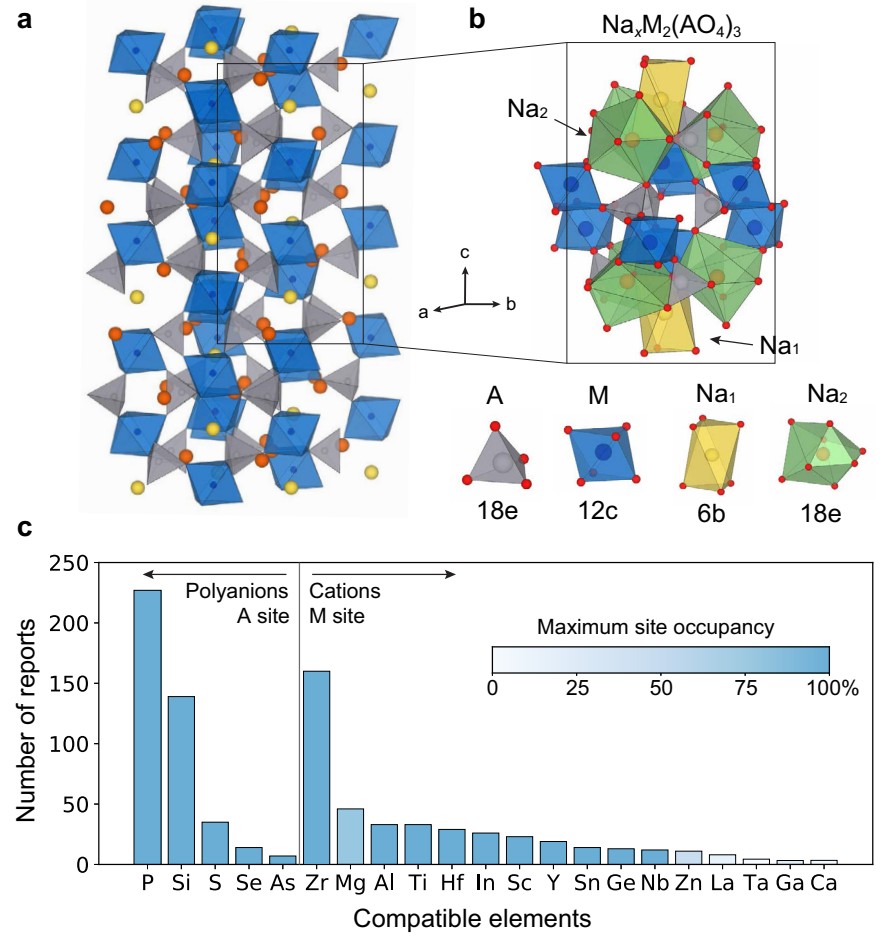

**Fig. 1 Structure and chemical space of NASICON. a** A Rhombohedral unit cell of NASICON-structured $Na_4M_2(AO_4)_3$ with blue units representing $MO_6$ octahedras, gray $AO_4$ tetrahedras, and yellow/orange spheres Na ions. **b** Local structure showing distinct cation sites. **c** The number of reports of NASICONs containing a specific element. Compounds are either grouped by the central cation A in the polyanion group (left), or by the redox-inactive metal M (right). The color bar shows the maximum observed site occupancy of each element in the literature.

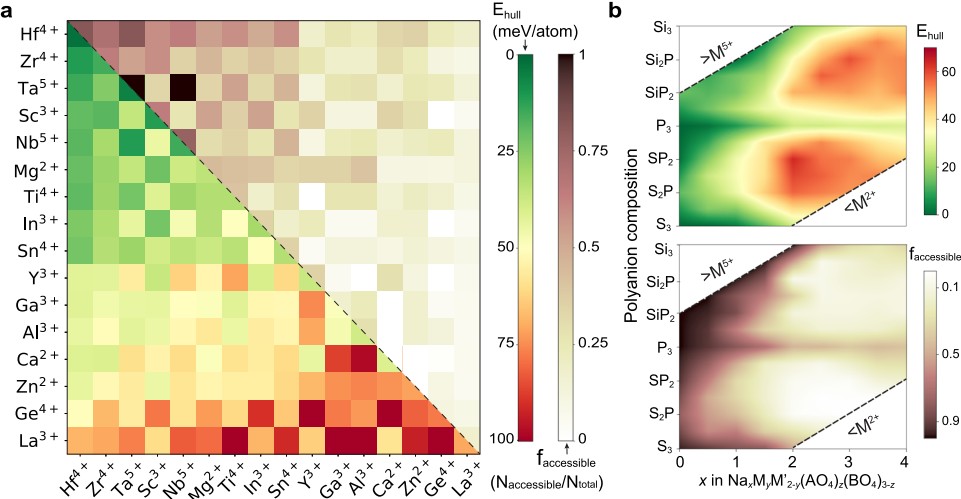

**Fig. 2 The thermodynamic stability of 3881 calculated NASICONs. a** Stability map. The color of each block in the lower triangle represents the median $E_{hull}$ (bottom-left) of NASICONs that contain the metals on the $x$- and $y$-axis. The upper triangle shows the fraction of compounds ($f_{accessible}$) with $E_{hull} - S_{ideal} \times 1000K \leq 0$. **b** The distribution of $E_{hull}$ (top) and $f_{accessible}$ (bottom) as a function of Na content and polyanion composition. Polyanions are represented by their central cation, e.g., $Si_2P$ indicates $(SiO_4)_2(PO_4)$. Green (red) indicates good (poor) NASICON stability in **a** and **b**.

**Table 1 Statistics of calculated NASICONs.**

| Materials category | Number/All | Percentage |
|---|---|---|
| Thermodynamic ground state (GS) ($E_{hull} \leq 0$) | 245/3881 | 6.3% |
| Likely synthesizable (LS) ($0 < E_{hull} \leq S_{ideal} \times 1000 K$) | 396/3881 | 10.2% |
| Unlikely synthesizable (US) ($E_{hull}$å $S_{ideal} \times 1000 K$) | 3240/3881 | 83.4% |
| Distribution of GS/LS-NASICONs | | |
| $3 \leq N_{Na} \leq 4$ | 60/1199 | 5.0% |
| $1 < N_{Na} < 3$ | 127/1576 | 8.1% |
| $N_{Na} \leq 1$ | 454/1106 | 41.0% |
| Pure phosphate/silicate/sulfate | 225/683 | 32.9% |
| Mixed polyanions | 416/3198 | 13.0% |

All calculated NASICONs are separated into three categories: thermodynamic ground states (GS) ($E_{hull} \leq 0$), likely synthesizable (LS) ($0 < E_{hull} \leq S_{ideal} \times 1000K$) and unlikely synthesizable (US) ($E_{hull} > S_{ideal} \times 1000K$).

with other metals, whereas $Ca^{2+}$, $Zn^{2+}$, $Ge^{4+}$, and $La^{3+}$ are metal species that are difficult to stabilize in a NASICON structure. Moreover, as indicated by the green (dark) region in the upper (bottom) panel of Fig. 2b, NASICONs are relatively more stable with low Na content or with a single type of polyanion.

To obtain more statistical information about the distribution of $E_{hull}$ among the calculated compositions, all NASICONs are separated into three categories: thermodynamic ground states (GS) ($E_{hull} \leq 0$), likely synthesizable (LS) ($0 < E_{hull} \leq S_{ideal} \times 1000K$) and unlikely synthesizable (US) ($E_{hull} > S_{ideal} \times 1000K$). The fractions of the three NASICON groups are shown in Table 1, with the GS- and LS-NASICONs only making up 6.3 and 10.2% of the computed compositions, respectively. The distribution of GS/LS-NASICONs across Na content and polyanion compositions are also shown in Table 1. In total, 41.0% of the NASICONs with $N_{Na} \leq 1$ are in the GS/LS group, which is more than eight times the GS/LS percentage in the $3 \leq N_{Na} \leq 4$ region. In addition, the percentage of GS/LS-NASICONs with a single type of polyanion is three times higher than that of mixed polyanions NASICONs.

**Experimental validation of the predicted Na-rich mixed-polyanion NASICONs.** To identify which NASICON compositions may have a higher probability of being interesting fast-ion conductors, we extracted the experimentally measured conductivity vs. Na content $x$ for more than two hundred NASICONs from the literature. The results are plotted in Fig. 3a (all conductivity values and literatures are tabulated in supplementary note 4). In general, the conductivity increases with Na content until $x \approx 3$. Therefore, we focus our synthesis effort on novel NASICONs with Na-rich stoichiometry (Na content $x \geq 3$).

The top panel of Fig. 3b shows the number of predicted GS/LS-NASICONs containing a specific M-M′ pair with Na content $x \geq 3$. As compared to the bottom panel, which shows similar information for experimentally reported NASICONs, it appears to have many stable compositions beyond the well-known Zr-containing NASICONs[29]. Therefore, there is still a large compositional space that remains undiscovered, within which 60 GS/LS compounds are predicted (Table 1). To further narrow down the target for experimental validation, we focused on the $Na_3$-NASICONs with mixed $SiO_4$ and $PO_4$ polyanions because such NASICON compositions are 1) more likely to be superionic conductors[46] and 2) more challenging to synthesize as they have both high Na content and mix polyanions (Fig. 2b, Table 1) which makes them good candidates to test the accuracy of our prediction. This selection criterion narrows the synthesis attempts

to the six NASICONs highlighted with a red rectangle in the bottom panel of Fig. 3b.

Figure 3c shows the X-ray diffraction patterns of the six synthesized NASICON samples. While five out of the six target NASICONs are obtained phase-pure or with only a trace amount of impurity phase, in the $Na_3HfSn(SiO_4)_2(PO_4)$ sample a NASICON phase coexists with a large fraction of $SnO_2$ impurity, indicating that the solubility of Sn may be limited under the synthesis conditions attempted. However, the presence of a NASICON phase in the diffraction pattern suggests that this composition may be synthesizable under different conditions (i.e., higher temperature, longer calcination time) or by an alternative synthesis route[47,48]. The high success rate of the experimental synthesis suggests that $E_{hull}$ values can be a good measurement of the synthetic accessibility, consistent with previous work[45].

**Physical trends of stability in NASICON.** The stability of the NASICON structure across the compositional space can be rationalized by the concept of bond compatibility and site miscibility. As schematically illustrated in Fig. 4a, there are two bond complexes in a NASICON structure: 1) the M-O-A bond complex, where the metal (M) and the central cation in the polyanion (A) compete for the covalency with oxygen; and 2) the M-O-Na bond complex, in which the electropositive nature of $Na^+$ results in largely localized electrons within the M-O bond. Because NASICONs can exist with mixed species on both the metal site (M) and polyanion central site (A), site miscibility (e.g., bond distortion due to size effect) is an additional factor influencing the stability.

Three types of electronegativity are considered to capture the bonding compatibility: the average electronegativity of the metal site $X_M^{Avg}$, the average electronegativity of the central cation in the polyanion $X_A^{Avg}$, and the electronegativity of sodium $X_{Na}$. The distribution of $X_M^{Avg} - X_{Na}$ and $X_A^{Avg} - X_M^{Avg}$ for all calculated NASICONs are plotted in Fig. 4b. The orange triangles mark all the computed NASICONs, while the blue lines define contours along which the fraction of GS/LS-NASICONs is constant, with higher fraction of stable NASICONs toward the center of the contours. Two groups of NASICONs are highlighted in Fig. 4b to illustrate the strong correlation between the electronegativity and phase stability. The first group contains three NASICONs with a stoichiometry of $Na_1$, i.e., A ($NaHfMg(SO_4)_2(PO_4)$), A1 ($NaHfCa(SO_4)_2(PO_4)$) and A2 ($NaGeMg(SO_4)_2(PO_4)$); while the second group contains three NASICONs with a stoichiometry of $Na_3$, i.e., B ($Na_3HfZr(SiO_4)_2(PO_4)$), B1 ($Na_3HfGe(SiO_4)_2(PO_4)$), and B2 ($Na_3Ge_2(SiO_4)_2(PO_4)$). Compositions A and B are representatives of NASICONs with the optimal metal electronegativity for $Na_1$ stoichiometry and $Na_3$ stoichiometry, respectively, both of which appear at a relatively central position of the contour lines in Fig. 4b. A1/A2 and B1/B2 are obtained by varying the electronegativities of the metal sites. In the first comparison group, the metal site is made more electropositive as Mg (A) is replaced by Ca (A1) (A→A1), while it becomes more electronegative with a substitution of Hf (A) to Ge (A2) (A→A2). Both substitutions increase the $E_{hull}$ value. More specifically, A→A1 increases $E_{hull}$ from 1.0 to 48.7 meV per atom, while A→A2 increases $E_{hull}$ from 1.0 to 54.8 meV per atom. In the second group, substitution of Zr by Ge (B→B1), or both Hf and Zr by Ge (B→B2) makes the metal site more electronegative. As a result, $E_{hull}$ rises from 4.0 meV per atom in B, to 56.0 and 90.3 meV per atom in B1 and B2. These results show that a deviation from the optimal electronegativity in the metal site leads to decreased NASICON stability.

To understand the electronic origin of the destabilization when moving away from the ideal electronegativity, the

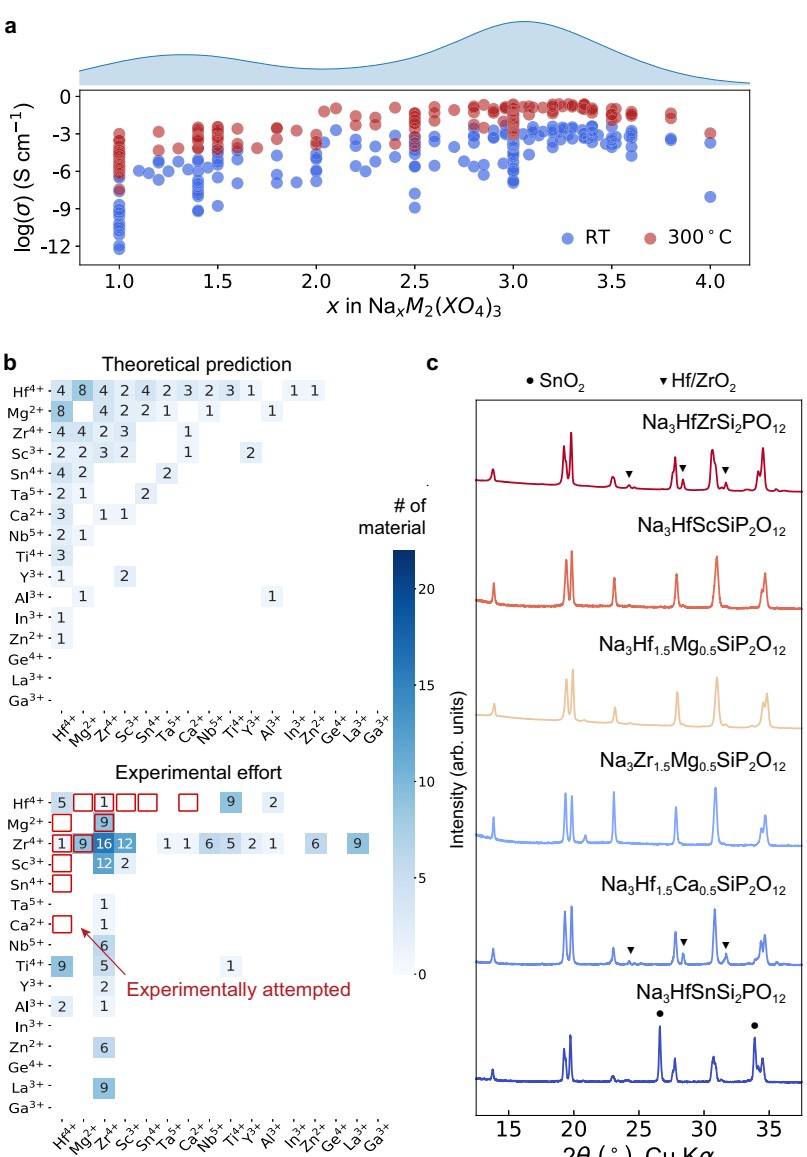

**Fig. 3 Solid-state synthesis exploration of predicted NASICONs within Na₃MᵧM'₁₋ᵧSizP₃₋zO₁₂ subspace.** **a** Experimentally measured conductivity in log scale vs. Na content of NASICONs at room temperature and 300 °C as extracted from the experimental literature. The upper curve shows the relative distribution of experimentally reported NASICONs as a function of Na content. A full list of conductivity data is presented in Supplementary Note 4. **b** The distribution of DFT predicted Na-rich GS/LS-NASICONs (top panel) and experimentally reported Na-rich NASICONs (bottom panel). The six experimentally attempted compounds in this work are highlighted with a red box. The number of NASICONs in each block is indicated by the color bar and specified by the number. **c** X-ray diffraction patterns of the six experimentally attempted NASICON compounds, five of which are pure NASICON phase or with a trace amount of impurity. The impurity phases are denoted with triangle ((Hf/Zr)O₂) and circle (SnO₂), respectively.

-ICOHP (Integrated Crystal Orbital Hamilton Population)[49–51] values of M-O bonds were calculated. As shown in Fig. 4c, with cation substitution A→A1, A→A2, or B→B1→B2, all the -ICOHP values drop, indicating weakened M-O bonds and thus higher $E_{hull}$ values as mentioned above. It is worth noting that the calculated -ICOHP values of A-O bonds in the polyanion group do not change much upon cation substitution (supplementary note 5), which can be explained by the strong covalency within the tetrahedrl polyanion group. The ICOHP data strengthens the idea that the NASICONs are stabilized by bond compatibility, as illustrated by the cartoon in Fig. 4a: the species in the M site should bond to O with reasonable covalency (setting a lower limit to the electronegativity) to stabilize the polarized M-O-Na complex; however, M-O cannot be too covalent (setting an upper limit for electronegativity)

otherwise it is not compatible with the A-O bonds in the polyanion group.

The size difference of the M cations will also influence the phase stability. In general, a large variation in size is detrimental to the phase stability[52–54]. In order to understand the influence of site miscibility, the distribution of NASICONs as a function of the bond length deviations extracted from the DFT computation are analyzed in Fig. 4d. The variations of the bond lengths $d$, $(d_{MO}^{max} − d_{MO}^{min})/d_{MO}^{max}$, are denoted as $\varepsilon_{MO}^{Variation}$ and $\varepsilon_{AO}^{Variation}$ for M-O and A-O bond, respectively. Similar to Fig. 4b, the fraction of GS/LS-NASICONs are plotted with contour lines. It can be observed in Fig. 4d that GS/LS-NASICONs are mostly found in the region with less bond deviations, consistent with the general idea that the size difference of ions mixed in a specific site contributes a positive amount of energy (favoring unmixing) to

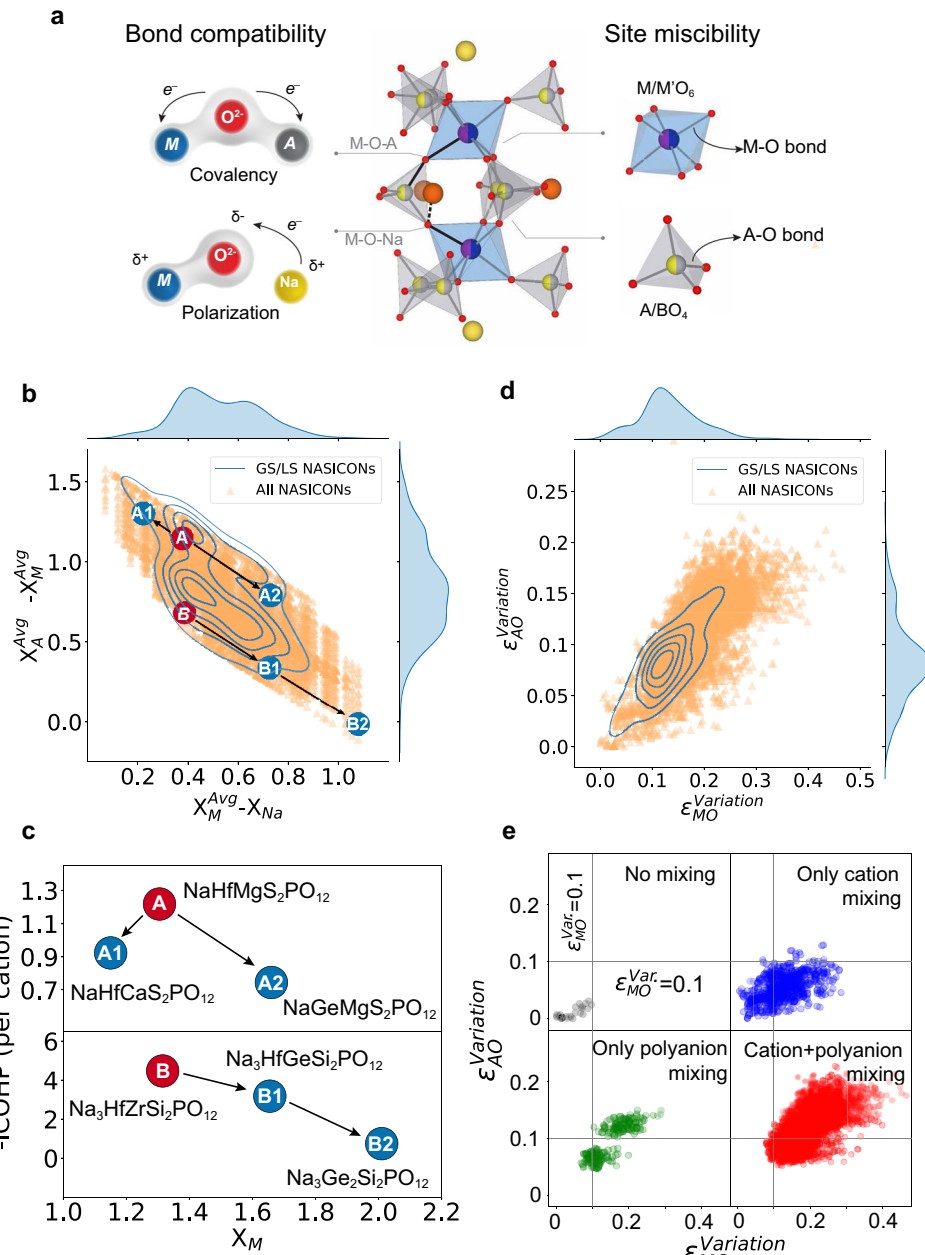

**Fig. 4 Physical factors that influence NASICON stability. a** Schematic of two main factors responsible for NASICON stability, i.e., bond compatibility and site miscibility. **b** Distribution of the computed NASICON compositions (orange triangles) in the space defined by two relevant electronegativity differences. x-axis: the difference between the average electronegativity of metal ($X_M^{Avg}$) and Na ($X_{Na}$); y-axis: the difference between the average electronegativity of metal ($X_M^{Avg}$) and polyanion central cation ($X_A^{Avg}$). The probability distribution of GS/LS-NASICONs are plotted with contour lines. The contours toward the center are of higher probability. Two groups of representative NASICONs are highlighted, i.e., A (NaHfMg(SO₄)₂(PO₄)) ($E_{hull}$ = 1.0 meV per atom), A1 (NaHfCa(SO₄)₂(PO₄)) ($E_{hull}$ = 48.7 meV per atom), A2 (NaGeMg(SO₄)₂(PO₄)) ($E_{hull}$ = 54.8 meV per atom). The other group contains three Na-rich NASICONs, i.e., B (Na₃HfZr(SiO₄)₂(PO₄)) ($E_{hull}$ = 4.0 meV per atom), B1 (Na₃HfGe(SiO₄)₂(PO₄)) ($E_{hull}$ = 56.0 meV per atom), B2 (Na₃Ge₂(SiO₄)₂(PO₄)) ($E_{hull}$ = 90.3 meV per atom); **c** The corresponding Integrated Crystal Orbital Hamilton Population (-ICOHP) per metal for M-O bonds of two groups in **b**. **d** The distribution of NASICONs as a function of metal-oxygen bond deviation ($\varepsilon_{MO}^{Variation}$) and central cation-oxygen bond deviation ($\varepsilon_{AO}^{Variation}$) in polyanions; The color code applied in all NASICONs and GS/LS-NASICONs are the same as **b**. **e** The distribution of bond deviations $\varepsilon_{MO}^{Variation}$ and $\varepsilon_{AO}^{Variation}$ for four groups of NASICONs, i.e., NASICONs with no mixing, with only cation mixing, with only anion mixing and with both mixing of cations and anions.

the energy of formation[52–54]. The mixing effects introduced by cation and polyanion are further distinguished and plotted in Fig. 4e. Clearly, the maximum lattice distortion is observed when both different cations and different polyanions are mixed. With only one mixed species, the polyanion mixing generally introduces a larger $\varepsilon_{MO}^{Variation}$ and $\varepsilon_{AO}^{Variation}$ than when M cations

are mixed. This explains the more pronounced destabilization of mixed-polyanion NASICONs compared to mixed metal NASI-CONs observed in Fig. 2. It is possible that the A-O bond is less tolerant to length variation and therefore has to pass size effects on as distortions in the M-O bonds. The ability of transition metals, in particular those with a $d^0$ configuration, to be

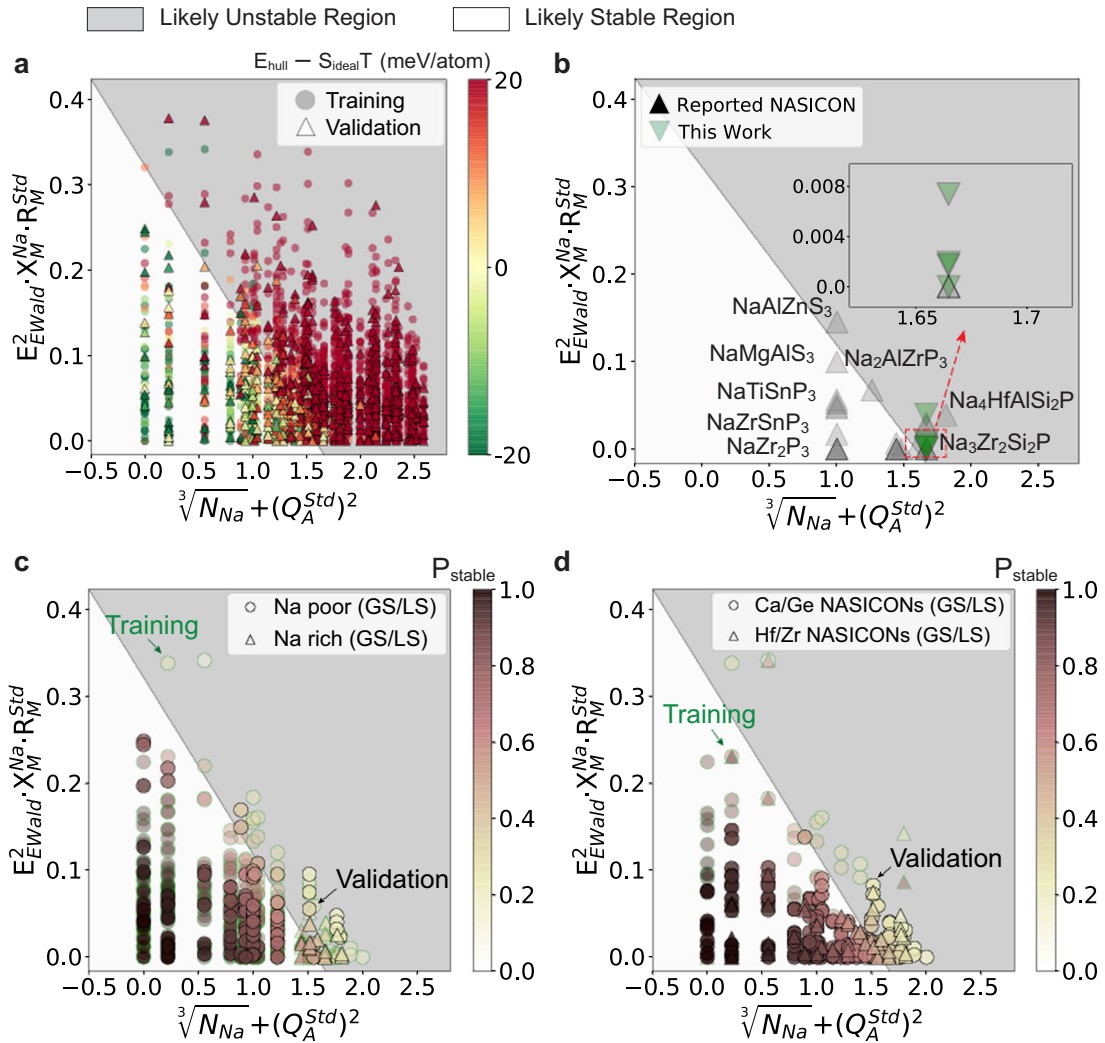

**Fig. 5 The distribution of different groups of NASICONs in the machine-learned feature space.** The white and gray shaded regions in each plot are SVC predicted spaces for GS/LS- and US-NASICONs, respectively. The symbols in axis labels represent the cube root of Na contents ($\sqrt[3]{N_{Na}}$), standard deviation of anion charge state ($Q_A^{Std}$), square of Ewald summation ($E_{Ewald}^2$), electronegativity difference between average cation electronegativity and Na electronegativity ($X_M^{Na}$) and standard deviation of cation radius ($R_M^{Std}$). **a** The Distribution of all NASICONs in the machine-learned 2D space. The computed $E_{hull}$ - $S_{ideal}T$ is represented by the color of the data point (T is set as 1000 K). Data points in the training (circles) and validation set (triangles) are plotted separately. **b** Reported and newly synthesized NASICONs plotted in the machine-learned 2D space. Inset shows a zoom-in area of newly synthesized NASICONs; **c** Na-rich ($Na_3$ per f.u.) and Na-poor ($Na_1$ per f.u.) GS/LS-NASICONs plotted in the machine-learned 2D feature space, with the accessible probability predicted by SVC indicated by the color of the data point. Data points in the training and validation set are plotted with green and black marker edge color, respectively. **d** Ca/Ge or Zr/Hf containing GS/LS-NASICONs in the machine-learned 2D space, with the accessible probability predicted by SVC indicated by the color of the data point. Data points in the training and validation set are plotted with green and black marker edge color, respectively.

accommodated in distorted octahedra has recently been explained[55].

**Machine-learned tolerance factor for NASICONs.** Since bond compatibility and site miscibility are two major physical factors responsible for the stability of NASICONs, a useful phenomenological model that can capture and quantify those physical factors would further facilitate the search of new NASICON compositions. To create a model that embeds the important physical information, while retaining enough simplicity for interpretation and analysis, we combined sure independence screening (SIS)[40–42] and machine-learned ranking (MLR)[43,44] approaches to search for the best mathematical representation of NASICON stability[42,56]. For easy visualization the model is limited to two dimensions (i.e., two fitting coefficients in the final model). Starting with 24 basic physical properties (primary

features) including elemental, compositional and structural information of NASICONs, we iteratively applied 17 mathematical operators to generate features set with 1,895,020 candidate features (details provided in the methods and supplementary note 6). SIS was then performed to select the top 2000 1D features that have the highest correlation with the $E_{hull}$ values. After that, we performed MLR to identify the best combination of two features for ranking the relative stability, i.e. the ordering of the $E_{hull}$ – $S_{ideal}T$ values for all the compositions. Our SIS+MLR process can be regarded as a variation of the SISSO method developed by Ouyang et. al.,[42] except that in our case MLR optimizes the ranking of relative stability, i.e., the ordering rather than the absolute value of $E_{hull}$- $S_{ideal}T$. Proper ranking of energies is more useful than focusing on the accuracy of the absolute energy to evaluate the synthesizability of materials[10]. The optimal 2D descriptor was chosen as the one that best validates the accuracy of the pairwise ranking matrix (the fraction of the validation set

correctly ranked by the descriptor, details in method), which is 84.7% on the 20% validation data from stratified sampling with a F1 score of 72.3%.

The distribution of the computed NASICONs in the SIS+MLR-learned 2D descriptor space is plotted in Fig. 5a, with the $E_{hull} - S_{ideal}T$ value of each compound indicated by the color bar. The first dimension of the descriptor is formulated as $t1 = \sqrt[3]{N_{Na}} + (Q_A^{Std})^2$, where $N_{Na}$ is the Na content and $Q_A^{Std}$ is the standard deviation of the electronegativity of the central cation in polyanions. The second dimension of the descriptor is formulated as $t2 = E_{Ewald}^2 \bullet X_M^{Na} \bullet R_M^{Std}$, where $E_{Ewald}$ is the Ewald energy of the lattice model normalized by that of $Na_3Zr_2(SiO_4)_2(PO_4)$ (details in supplementary note 6); $X_M^{Na}$ the metal site electronegativity taking the Na electronegativity as a reference, and $R_M^{Std}$ the standard deviation of the M site cation radius normalized by the ionic radius of $Na^+$. The first dimension essentially captures the Na content and polyanion mixing effect, while the second dimension captures the cation mixing effect, both of which are consistent with the important physical factors identified in Fig. 4. It should also be noted that we used a dimensionless $E_{Ewald}$ and $R_M^{Std}$ to keep the MLR and later classification process physically meaningful. In Fig. 5a, most of the US-NASICONs are well separated from the GS/LS-NASICONs in the 2D descriptor space, and a linear decision boundary can be drawn by the support vector classification (SVC) with a stratified fivefold cross-validation (details in methods). In Fig. 5a, the compounds used for the training model and the validation model are denoted with different marker shapes. While the average validation accuracy and macro F1 score (methods) of the fivefold cross-validation are 82.1 and 74.7%, respectively, the highest performance model has a validation accuracy of 84.5% and F1 score of 78.1%, which does not differ much from the average model, indicating that the model is not overfitted. Moreover, the average training accuracy (82.3%) and F1 (74.9%) are close to the validation metrics, further indicating that overfitting is unlikely. Detailed metrics can be found in methods and supplementary note 7. Out of the 5 validation sets, the model with a macro F1 score closest to the average is selected as the representative, which is plotted in Fig. 5a, and used in the following discussion. With the selected linear SVC model, a synthetically accessible NASICON at 1000 K should satisfy the condition: $0.203 \times t1 + t2 \leq 0.322$.

To evaluate the model performance, the selected SVC boundary was first applied to classify 27 previously reported and five newly synthesized NASICONs (Fig. 5b). For each compound, the Platt scaling probability[57] ($P_{stable}$) calculated from the trained SVC model is used to estimate the probability of being synthetic accessible. When compounds with $P_{stable} \geq 50\%$ are classified as synthetic accessible, the stability model correctly captures 18 out of 32 experimentally synthesized NASICONs. The 56.3% recall value is obviously lower than the validation recall value on LS/GS NASICONs (81.3%, details in supplementary note 7), which can be regarded as performance variation in different data domains considering that the experimental data may not be uniformly distributed. Specifically, the 32 experimental compositions contain 12 Na-poor ($Na_1$ per formula unit, f.u.) and 18 Na-rich NASICONs. While the recall value for the Na-poor systems is 11/12, the model only performs with a low recall value of 6/18 for the Na-rich compounds. Meanwhile, all the five newly synthesized $Na_3$ NASICONs are misclassified. We do not think this is due to the overfitting of our SVC model because the fivefold cross-validation gives very similar validation error (supplementary note 7). However, such a behavior indicates that the performance of the SVC decision boundary indeed varies with composition. It is also worth mentioning that 8 of the 14 false-negative predictions ($P_{stable} < 50\%$, but experimentally stable) have $\geq 40\%$ probability to be accessible, and even the worst false-negative has a $P_{stable}$ value of 18.5% (supplementary

note 8). For the five newly synthesized $Na_3$-NASICONs, 4 of them have a $P_{stable}$ close to 40%, while the last one, $Na_3Ca_{0.5}Hf_{1.5}(SiO_4)(PO_4)_2$, has a $P_{stable}$ of 27.4%. Comparing to the $P_{stable}$ distribution of US-NASICONs (supplementary note 9), those false-negative predictions still sit much closer to the decision boundary that most unstable NASICONs. This observation implies that though our simple model shows weakness in the prediction of Na-rich NASICONs, the prediction is still useful for guiding experimental synthesis.

Since the model exhibits different performance for different Na compositions, we divided the data into different compositional groups based on 1) Na content and 2) metal chemistry, and investigated the model performance on Na-rich/Na-poor NASICONs, as well as on NASICONs with different metal chemistry (Ca/Ge vs. Hf/Zr).

The distribution of GS/LS-NASICONS with $Na_1$ per f.u. and $Na_3$ per f.u. and corresponding accessible probabilities ($P_{stable}$, indicated by the colormap) predicted by the above-mentioned SVC model are shown in Fig. 5c (the distributions of all $Na_1$ and $Na_3$ NASICONs are shown in supplementary note 10). We separate the compounds used for the training model and validation model with different marker edge color. The validation recall values for classifying Na-poor and Na-rich NASICONs are 87.9% and 25%, respectively, while the corresponding training recall value is 93.3% for Na-poor NASICONs and 25% for Na-rich NASICONs, which again confirms that Na-poor NASICONs are better captured by this model compared to Na-rich ones. This performance limitation likely originates from the fact that we are trying to optimize the performance of the model in a global compositional space, making the performance in the Na-rich subspace not optimal. Moreover, the average $P_{stable}$ of false-negative predictions is 30.4% for $Na_1$ NASICONs and 37.7% for $Na_3$ NASICONs, which are still larger than most of the US-NASICONs as shown in supplementary note 9.

Similarly, the distribution of GS/LS-NASICONs containing Ca/Ge or Zr/Hf metal species are plotted in Fig. 5d (the distribution of all Ca/Ge or Zr/Hf containing NASICONs is plotted in supplementary note 10). NASICONs with Hf/Zr species are often predicted to be GS/LS (Fig. 2), thereby serving as a good contrast to the Ca/Ge group which has a much lower GS/LS frequency (Fig. 2). The recall values for Ca/Ge and Hf/Zr NASICONs in the validation set are 72.7% and 84.2% respectively, which are quite comparable, indicating that there is little dependency of the prediction ability on metal chemistry. Additionally, the false-negative points are not far off the decision boundary, as the average probability of being accessible is 34.0% for the Hf/Zr group and 34.3% for Ca/Ge group. It can therefore be concluded that our machine-learned stability model should be able to offer reasonably stability prediction for NASICONs regardless of their metal chemistries.

## Discussion

The efficient discovery of inorganic solids with complexity in both composition and crystal structure is a challenging task. While simple modifications to a basic compound are prevalent in the literature, as is the case for NASICONs[35,46,58,59], there is a lack of thorough understanding of the stability rules to help one navigate through the unexplored chemical space. In this work, we conducted high-throughput phase diagram calculations to evaluate 3881 possible NASICON compositions. We not only successfully synthesized five out of six predicted $Na_3$ mixed-polyanion NASICONs but also rationalized the variation in thermodynamic stability as a function of Na content, metal species, and polyanion mixing. Our analyses together with a

machine-learned predictive phenomenological model points out several key factors for NASICON stability:

1. The electronegativity of a metal is bounded on both sides for it to be stable in a NASICON structure due to its covalency competition with the cation in the polyanion and the necessity to maintain Na ionicity. A metal that is too electronegative favors competing phases where those metals are present in a more covalent bonding environment. To give one example, Ge-containing NASICONs, such as $Na_3HfGe(SiO_4)_2(PO_4)$ (B1) and $Na_3Ge_2(SiO_4)_2(PO_4)$ (B2), tend to decompose into other phases with the Ge accommodated in $Na_4Ge_9O_{20}$. The calculated -ICOHP per Ge-O bond is 1.65 and 1.38 in B1 and B2 compared with 3.38 in $Na_4Ge_9O_{20}$ (details in Supplementary Note 11), which supports the strengthening of Ge-O bond upon decomposition. On the other hand, a metal that is too electropositive metal is also unstable in NASICON, as demonstrated by the Ca-containing NASICONs such as $NaHfCa(SO_4)_2(PO_4)$ (A1). The small -ICOHP per Ca-O bond, 0.22, indicates a low covalency of the Ca-O bond, and its decomposition into $CaSO_4$ removes the Na-Ca competition for ionization and enhances the strength of Ca-O bond (-ICOHP = 0.64 per bond) (details in Supplementary Note 12). This concept of bond competition can be extended to other polyanionic ceramic materials that contain multiple metal sites. One important group of such materials is the alkali (earth) containing polyanion materials, which are widely studied as both electrodes and solid electrolytes in the energy storage field[59–61]. Those materials generally have two metal sites, one of which is the alkali (earth) metal site, while the other is typically a transition metal site whose polyhedron corner-shares with polyanions. In such materials, the alkali (earth) metal site would set the lower bound of electronegativity, while the upper bound is set by the central cation of polyanion in order to maintain the strong covalency within the polyanion group.

2. The size difference of the metal and polyanion cation also influences the stability. The bond length variation analysis (Fig. 4d) demonstrates that the polyanion site is generally less tolerant to lattice distortion. Therefore, most of the misfit strain due to mixing polyanions has to be passed on to the cation site. If that is not sufficiently possible, the compound will form other phases to release the strain energy. Our analysis on NASICON materials thus explains why it is in general more difficult to create mixed-polyanion systems.

Our work also demonstrates a computational paradigm to evaluate the stability of complex ceramic materials with phenomenological models derived from machine learning. Within this framework, we use $E_{hull}$ and ideal mixing entropy model to estimate the synthetic accessibility of materials at a finite temperature. With such a quantification of the synthetic accessibility, machine learning framework can then be applied to rank the $E_{hull}$ values, which thus will be useful for predicting synthetic accessibility.

The complicated compositional variety and bond topology of many inorganic materials makes the development of a phenomenological stability model similar to Goldschmidt's tolerance factor remarkably difficult. However, a relatively simple and quantitatively predictive model was obtained in this work by applying the sure independence screening combined with machine learning ranking algorithms to search through billions of possible mathematical representations. The successful derivation of a tolerance factor for NASICON provides an example of how effective such a toolset can be for establishing stability rules for complex oxides starting from a certain amount of DFT data. Once a simple machine-learned model is established, no additional DFT calculations are required for exploring the broader chemical space. Such a framework would be remarkably useful for multicomponent ceramic materials with complex structures[62,63].

## Methods

**Experiments**. For the solid-state synthesis of NASICON compounds, typical metal oxides or hydroxides ($HfO_2$ (Aldrich, 99.8%), MgO (Aldrich, ≥99.99%), $Sc_2O_3$ (Sigma, 99.9%), $Zr(OH)_4$ (Aldrich, 97%), $SnO_2$ (Alfa Aesar, 99.9%), CaO (Sigma-Aldrich, 99.9%)) were used as precursors to introduce metal cations. $SiO_2$ (Sigma-Aldrich, nanopowder) and $NaH_2PO_4$ (Sigma, >99%) were used as silicate and phosphate sources. $Na_2CO_3$ (Sigma-Aldrich, >99%) was used as an extra sodium source. In addition, 10% excess $NaH_2PO_4$ was introduced to compensate for the possible sodium and phosphate loss during the high-temperature treatment. The powder mixtures were wet ball-milled for 12 h using a Planetary Ball Mill PM200 (Retsch) to achieve a thorough mixing before pressed into pellets. The pelletized samples were first annealed at 900 °C under Ar flow, then grounded with mortar and pestle, wet ball-milled, pelletized, and re-annealed at 1100 °C. The crystal structures of the obtained materials were analyzed using X-ray diffraction (Rigaku Miniflex 600 and Bruker D8 Diffractometer) with Cu Kα radiation.

**First-principles calculations**. First-principles total energies calculations were performed with the Vienna ab initio simulation package (VASP) with a plane-wave basis set[64]. Projector augmented-wave potentials[65] with a kinetic energy cutoff of 520 eV and the exchange-correlation form in the Perdew–Burke–Ernzerhof generalized gradient approximation (GGA-PBE)[65] were employed in all the structural optimizations and total-energy calculations. For all the calculations, a reciprocal space discretization of 25 k-points per $Å^{-1}$ was applied, and the convergence criteria were set as $10^{-6}$ eV for electronic iterations and 0.02 eV/Å for ionic iterations. A rhombohedral conventional cell was used for each of the NASICON structures, and the Na-vacancy ordering, cation ordering, and anion ordering were set as the one with lowest electrostatic energy to approximate the equilibrium ionic ordering.[59,66–69] It should be noted that NASICONs can also exist with other space group, such as the monoclinic form (C2/c). The monoclinic NASICONs can be regarded as an ordered version of rhombohedral NASICONs[70]. Therefore, the energy difference between rhombohedral and monoclinic NASICONs is expected to be low as it only comes from Na disordering energy. To give an estimation, the disordering energy of Na site is typically 30–40 meV, while the contribution to $E_{hull}$ has to be normalized by the number of atoms/f.u., i.e., ~30–40 meV/ (17–21 atoms/f.u.). Such energy variation is much smaller compared with the energy variation caused by chemistry variation. A table comparing the energetics between 15 groups of Rhombohedral and Monoclinic NASICONs are also attached in supplementary note 3 to support the argument. With such considerations, all our calculations and analysis are only on rhombohedral NASICONs data for simplicity. In addition to the 3881 NASICON structures, all the competing phases in the relevant chemical spaces that are given in The Materials Project[1] were also calculated to construct the phase diagrams. To estimate the synthetic accessibility of the NASICON at finite temperature, we assumed that a synthetic accessible NASICON could be made due to a configurational entropy stabilization, i.e., $E_{hull} - TS_{Ideal} \leq 0$. Therefore, the ideal entropy was calculated by assuming a fully disordered distribution of Na, cation, and anion sites (Supplementary Note 1).

**Machine-learning setup**. To consider as many physical factors as possible, 24 basic features covering the basic element properties of chemical species are constructed (a list of features is presented in supplementary note 6). Combination of these features with 17 mathematical operators listed in supplementary note 6 with a complexity of two leads to 1,895,020 possible features. SIS was then performed on 80% of the data, using the SISSO package developed by Ouyang et.al.[42] to identify the top 2000 features with the highest Pearson correlation to $E_{hull}$ values. More information of the SIS process can be found in Supplementary note 6.

3104 out of 3881 data points are generated from stratified sampling due to unbalanced sample amount between the GS/LS-NASICONs and US-NASICONs. With the selected 2000 features, the possible 2D models will be $C_{2000}^2 = 1,999,000$. A ranking support vector machine (SVM) model was trained to select the model with the best capability of ranking the $E_{hull}$ values, as demonstrated in Fig. 5. The objective function of the SVM is set as the accuracy of predicting the pairwise ranking matrix (i.e., a $3881 \times 3881$ matrix that describes the relative order of any pairs of compositions) for model optimization. During the MLR process, we used the stratified sampling training set applied at SIS step for optimizing the hyperparameters. The validation accuracy on the 20% computing data left was used as the target function to optimize through hyperparameter searching. More discussion and validation of the reliability of this method for avoid overfitting can be found in Supplementary Note 6.

With the SIS+MLR-learned 2D descriptor space, a linear decision boundary of the synthetic accessible NASICONs was drawn with the assistance of support vector classification (SVC) with balanced weight given that the populations of GS/LS samples are not equivalent. It should also be noted that all the F1 scores presented here are averaged F1 scores (macro F1 scores) for predicting both LS/GS NASICONs and US-NASICONs. We use the averaged F1 score because it can provide a better assessment of model performance for classification problems with imbalanced classes, as 83.4% of the computed NASICONs are US. Also due to the uneven distribution of LS/GS and US-NASICONs, we used class-weighted SVC, where the weight is inversely proportional to the amount of data in that class. A complete list of validation accuracies, recall values, and F1 scores are listed in Supplementary note 7. The model with F1 score closest to the average value during cross-validation is selected as the final model.

## Data availability

The phase stability data, machine learning model and code to reproduce our tolerance factor in this study has been deposited in a Github Repository[71] under access code https://github.com/Jeff-oakley/NASICON_Predictor_Data. The conductivity data used in Fig. 3a are available in Supplementary information. The data plotted in Figs. 1c and 3c are provided with the manuscript (raw_data.xlsx). Source data are provided with this paper.

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

## Acknowledgements

This work was supported by the Samsung Advanced Institute of Technology. The computational analysis was performed using computational resources sponsored by the Department of Energy's Office of Energy Efficiency and Renewable Energy at the National Renewable Energy Laboratory. Computational resources were also provided by the Extreme Science and Engineering Discovery Environment (XSEDE), which is supported by the National Science Foundation grant number ACI1053575 and the National Energy Research Scientific Computing Center (NERSC), a DOE Office of Science User Facility supported by the Office of Science and the U.S. Department of Energy under contract no. DE-AC02- 05CH11231. This research used the Lawrencium computational cluster resource provided by the IT Division at the Lawrence Berkeley National Laboratory (Supported by the Director, Office of Science, Office of Basic Energy Sciences, of the U.S. Department of Energy under Contract No. DE-AC02-05CH11231). The authors also thank Zijian Cai and Leeann Sun at University of California, Berkeley for their assistances during the NASICON synthesis.

## Author contributions

B.O., J.W. and G.C. initiated and designed the project. G.C. supervised all aspects of the research. B.O. performed the high-throughput DFT calculations. B.O. and J.W. analyzed the high-throughput data and conceived the phase stability trend. J.W. synthesized all proposed compounds. B.O. conducted the ICOHP calculations. B.O. performed the machine learning work with the help of T.H., H.H., C.J.B. and J.W. Y.W., V.L. and H.K. contributed valuable discussion and insights. B.O., J.W. and G.C. wrote the manuscript. The manuscript was revised by all authors. B.O. and J.W. contributed equally to this work.

## Competing interests

The authors declare no competing interests.
