## [Peer Review File · Nature Communications]

REVIEWER COMMENTS

Reviewer #1 (Remarks to the Author):

In this work, the authors developed the stability rules of Nasicon-structured materials. By applying machine learning to the ab-initio computed phase stability of 3881 potential Nasicons the authors extracted a simple two-dimensional descriptor that is extremely good at separating stable from unstable Nasicons. Below are some comments:

1. In Table 1, the criterion for the thermodynamically ground state Nasicons is $E_{\text{hull}} \leq 0$. The E_{hull} of the ground state materials is 0, what does " <0 " mean?
2. In exploring Nasicons, the authors considered different Na contents, different transition metal combinations and different polyanion combinations. How did they determine the positions of Na^+ , transition metals and polyanions in the unit cell?
3. In the machine learning part, the authors noted that "a useful phenomenological model that is able to capture those physical factors and quantify them without additional DFT calculations can further facilitate the search for new NASICON compositions". Is it necessary to optimize the initial structures by DFT calculations?
4. For Nasicons, there are many different crystal forms, such as rhombohedral and monoclinic. Could the method developed in this work be applied to determine the stability for other non-rhombohedral Nasicon-type materials?
5. Some latest reviews on NASICON solid electrolytes such as Mater. Today 2020, 41, 200-218 and ChemElectroChem DOI: 10.1002/celec.202001527 are recommended.

Reviewer #2 (Remarks to the Author):

The authors compute the formation energy of 3000+ NASICON materials with DFT, compute the phase diagrams to extract the energy above the convex hull, successfully synthesize 5 new materials, and then develop a two-descriptor model to describe their data.

Overall, I find the computational and experimental aspects of this work to be exciting and well-executed. The analysis of the structural factors influencing stability is similarly well done. However, I feel like the ML modeling side of the work is a bit muddled and introduces many questions. I think the work may be suitable for publication but would benefit from some clarifications and revisions before publication.

First, a few minor points:

- Without meaning any insult to the authors, some of the English should be improved for readability, particularly in the abstract and introduction.
- Krishna Rajan et al.'s 2011 work on predicting stability in perovskites should probably be cited along with the current references 12-17 (<https://doi.org/10.1098/rspa.2010.0543>)
- How can you have an $E_{\text{hull}} < 0$, as referenced in Table 1?
- The conductivities of over 300 NASICON materials are referenced in Fig. 3, but there is no indication given as to where these measurements came from. Please provide the data in the SI or cite the paper(s) where these data came from (or, ideally, both)

A few general/high-level comments and questions:

- It appears to me that the work is missing a compelling argument that computational predictions of the thermodynamic phase stability are actually correlated with experimental stability. Many metastable materials are synthesizable and stable at ambient conditions. This is a fairly big claim and

it is central to the author's work; can the authors reference past studies that have demonstrated convincingly that this is a safe assumption?

-- It is stated in the introduction that synthesizability is a non-local property, and yet the authors attempt to fit local descriptors to predict it. Can the authors address this apparent contradiction?

-- The "Physical trends of stability in NASICON" section is very good. My takeaway is that understanding the physical system in depth makes for promising descriptors, and that good descriptors are necessary for small data models. However these descriptors are very clearly not generalizable beyond the NASICON materials, despite some implications that these descriptors could provide value in other materials systems, e.g. in the introduction: "...but also demonstrate an efficient paradigm for discovering new materials with complicate composition and atomic structure". I don't think this necessarily a new paradigm that's being described here.

Finally, a few points on the ML section. To make a compelling case, the the authors first need to argue that the computational stability calculations are correlated with experimental stability (as mentioned above). Then, they need to show that the ML model is correlated with the computational data it was trained on. I'm not sure this is thoroughly shown.

-- The authors appear to quote training error when providing the model performance: "Applied to all 3881 compositions, such stability model has an overall classification accuracy of 81.8%, as well as a recall value of 96.6% and F1 score of 88.2% for capturing GS/LS-NASICONs." Was this training error? Validation? K-fold cross-validation? Training error should almost never be cited as a model generalization error proxy.

-- The authors examined the fit of ~4M models against 3K training points. With this many "attempts" to find the best model, one risks crossing the bounds dictated by PAC learning theory. Can the authors justify this approach within the bounds of PAC theory? Otherwise, overfitting may be an issue. Although the "test" set of 32 compounds is small, there is a hint of overfitting: 96.6% recall in training but 62.5% recall on the test set.

-- It appears the ML model made incorrect prediction(s) for the 5 successfully synthesized compounds according to Fig 5b. Can you confirm this? This would seem to challenge whether the ML model can actually predict experimental stability. I also only see two "new" materials on this plot; shouldn't there be 6?

-- The authors discuss how their model is validated against three criteria: "1) its ability to capture experimentally reported NASICONs (Fig. 5b); 2) its robustness in identifying GS/LS NASICONs in both Na rich and Na poor regions (Fig. 5c); 3) its robustness in identifying GS/LS-NASICONs with different metal chemistries (Fig. 5d)." However, actual performance metric values are not given for #2 or #3. Can you please clarify the model performance in these areas?

-- It is claimed a few times that the incorrect predictions are close to the decision boundary ("Moreover, the false negative classifications still remain likely to be synthetically accessible, as the average accessible probability of the false negative predictions is 30.9% for Na₁ NASICONs and 39.0% for Na₃ NASICONs.") However, this statement is hand-wavy without knowing the distribution of other points around the decision boundary and also begs the question: then why not move the decision boundary? I believe it is better to be forthright about the model performance without discussing how the model wasn't "too wrong."

Reviewer #3 (Remarks to the Author):

The manuscript by Bin Ouyang et al. present high-throughput computations for the stability and synthetic accessibility of new NASICON materials which have complex bonding topology and composition, followed with machine learning to develop a simple descriptor to describe the stability of NASICONs. Evaluation of stability and synthetic accessibility were made with the quantity of energy

above the convex hull (Ehull) taking configurational entropy into account, and reliability of the computed data were demonstrated by experimental confirmation on six predicted new materials. This work provides significant results in that not only a new stability map of NASICONs was developed from the high-throughput computational results which is of potential importance for practical applications, but also the physical trends of the NASICONs stability was understood with the concept of bond compatibility and site miscibility, and based on which a simple descriptor ("tolerance factor") for NASICONs was identified. While this work brings novelty in the discovery and understanding of the complicated systems NASICONs, additional evidences are needed for the machine learning of the tolerance factor before I can recommend publication. My comments and suggestions are provided below.

Major comments:

1. Convergence of the machine learning results is unclear. In this work, the SIS+MLR machine learning scheme were used. The "SIS" part involves the generation of increasing complex features by using the SISO code, and the complexity was simply set to two. Actually, the feature complexity (defined as the number of operators in a feature) is a hyper-parameter that need to be optimized. It is not clear whether there is a significant improve if the feature complexity is slightly increased up to 3 or 4. Similarly, it is not discussed why the SIS-selected features were set to 2000. What will be the results if this number is increased to 2500 or 3000? Or is it because the computational cost of the MLR part limit this number to only 2000?

2. In the first dimension of the 2D descriptor $t_1 = \sqrt[3]{(N_{Na}) + X_A^{Std}}$, the linear combination of Na content and standard deviation of electronegativity of A is physically meaningless, though the classification in Fig 5 looks good. N_{Na} is the number of Na atoms in the chemical formula $NaxM_2(AO_4)_3$, whereas the electronegativity X represents the tendency of an atom to attract electrons. It is difficult to understand why the linear combination of such two quantities is reasonable. I believe this is an important question the readers will ask if this descriptor is going to be impactful.

3. The prediction performance of the tolerance factor on unseen data is not clear. Fig 5 shows analysis of the descriptor with the training data, but I don't see a test on unseen data or cross validation (e.g. how sensitive is the descriptor on the training dataset?)

Minor comments:

1. The authors claim the descriptor is "extremely good" in the abstract. This seems overrated as the overall training classification accuracy is 81.8%, where there is still much room to improve.

2. More details are needed for the MLR methods and the primary features. For example, the definition and data source for the input features ionic radii (Shannon radii?) and electronegativity (Pauling electronegativity?) is unclear. In line 279, "MLR would optimize the ranking of relative stability, which is more important than the absolute Ehull values ...", please elaborate on the "relative stability" and why it is better than the absolute Ehull values.

Response to the Reviewers: Manuscript ID NCOMMS-21-06620

We thank the editor and three reviewers for their time spent to evaluate our manuscript and for their recommendation/comments. All of the reviewers' concerns are addressed below in detail. In addition to the modification in response to reviewer's comments, we have also made some other changes in language and technique contents. All changes to technique contents are colored in red, while language editing is colored in blue in the manuscript.

Reviewer #1 (Remarks to the Author):

In this work, the authors developed the stability rules of Nasicon-structured materials. By applying machine learning to the ab-initio computed phase stability of 3881 potential Nasicons the authors extracted a simple two-dimensional descriptor that is extremely good at separating stable from unstable Nasicons. Below are some comments:

1. In Table 1, the criterion for the thermodynamically ground state Nasicons is $E_{\text{hull}} \leq 0$. The E_{hull} of the ground state materials is 0, what does “<0” mean?

The E_{hull} is calculated as energy relative to a convex hull constructed from all the compounds available in Materials Project Data base (APL Materials 1, 011002 (2013), <https://materialsproject.org/>) without the newly computed NASICONs. Therefore, $E_{\text{hull}} \leq 0$ indicates that the evaluated NASICON is a new ground state. We have clarified this in the manuscript (Page 6 colored with red).

2. In exploring Nasicons, the authors considered different Na contents, different transition metal combinations and different polyanion combinations. How did they determine the positions of Na^+ , transition metals and polyanions in the unit cell?

Since there are almost 4000 NASICONs evaluated in this paper, we cannot evaluate all possible ion configurations with DFT. As a reasonable approach, we created a large number of distributions of Na on the 6b and 18e sites, polyanion distributions, and metal distributions in a Rhombohedral cell and ranked them according to their electrostatic energy. The configuration with lowest electrostatic energy was calculated with DFT. Similar sampling methods have also been applied in some of our previous work (Chem. Mater. 2020, 32, 5, 1896–1903, Adv. Energy Mater. 2020, 1903968, Adv. Funct. Mater. 2019, 1902392, Nature Chemistry, 2016, 8, 692–697, Chem. Mater. 2016, 28, 15, 5450–5460 etc.). Note that at this point we only need a reasonable estimate for the energy from this analysis to determine whether the NASICON is stable or not. So even if we do not have the exact occupancy from this, we are guaranteed a fairly low energy for each composition. We have modified the manuscript to clarify how we picked the ionic ordering (Page 6 colored with red).

3. In the machine learning part, the authors noted that “a useful phenomenological model that is able to capture those physical factors and quantify them without additional DFT

calculations can further facilitate the search for new NASICON compositions". Is it necessary to optimize the initial structures by DFT calculations?

The current model only requires basic physical properties such as cation radius, Na content, electronegativity and electrostatic energy. The first three parameters (i.e., cation radius, Na contents and electronegativity) are only related to the chemistry of the NASICON, while the fourth parameter (electrostatic energy) is calculated as the Coulomb interaction for a prototype NASICON framework with fixed lattice constant. None of these quantities requires DFT calculations. We have provided the prototype NASICON structure in our supplementary materials as well as an example python code demonstrating how to calculate the exact electrostatic energy terms on the basis of that structure (NASICONPrim.cif and CalculateEWaldEnergy.ipynb), relevant modifications are also added to the manuscript (Supplementary note 6 colored in red).

4. For Nasicons, there are many different crystal forms, such as rhombohedral and monoclinic. Could the method developed in this work be applied to determine the stability for other non-rhombohedral Nasicon-type materials?

We thank the reviewer for bringing this up. The major difference between the rhombohedral (R-3C) and monoclinic phases (C2/C) NASICONs is related to Na ordering (Chem. Mater. 2002, 14, 4684-4693). Therefore, the energy difference between rhombohedral and monoclinic NASICONs is expected to be small. To give an estimate, the disordering energy of Na site is typically 30-40 meV per f.u., while the contribution to E_{hull} has to be normalized by the number of atoms/f.u., i.e., $\sim 30-40$ meV/ (17-21 atoms/f.u.). Such energy variation is much smaller than the energy change caused by the chemistry of the material. A table comparing the E_{hull} values among 15 groups of NASICONs for both Rhombohedral and Monoclinic phases is also attached in supplementary note 3 to support the argument that the energy difference between both phases is small. Hence, the fact that we use the rhombohedral NASICON as a prototype should not change the conclusion as to whether a NASICON is stable or not at a given composition. But of course, we cannot conclude whether it is rhombohedral or monoclinic NASICON that will form. We have added clarification on this point in page 25 (colored red)

5. Some latest reviews on NASICON solid electrolytes such as Mater. Today 2020, 41, 200-218 and ChemElectroChem DOI: 10.1002/celc.202001527 are recommended.

We thank the reviewer for suggesting the review papers, we have included these papers into our references (Page 4 colored in red, literature 36 and 37).

Reviewer #2 (Remarks to the Author):

The authors compute the formation energy of 3000+ NASICON materials with DFT, compute the phase diagrams to extract the energy above the convex hull, successfully synthesize 5 new materials, and then develop a two-descriptor model to describe their data.

Overall, I find the computational and experimental aspects of this work to be exciting and well-executed. The analysis of the structural factors influencing stability is similarly well done. However, I feel like the ML modeling side of the work is a bit muddled and introduces many questions. I think the work may be suitable for publication but would benefit from some clarifications and revisions before publication.

First, a few minor points:

-- Without meaning any insult to the authors, some of the English should be improved for readability, particularly in the abstract and introduction.

We thank for the reviewer for pointing out the language issue. We have carefully gone through the manuscript and polished the language. We hope such efforts had improved the readability of our manuscript.

-- Krishna Rajan et al.'s 2011 work on predicting stability in perovskites should probably be cited along with the current references 12-17 (<https://doi.org/10.1098/rspa.2010.0543>)

We have added this work into our references (page 3 colored in red, literature 18)

-- How can you have an $E_{\text{hull}} < 0$, as referenced in Table 1?

See our answer to question 1 of Reviewer 1.

-- The conductivities of over 300 NASICON materials are referenced in Fig. 3, but there is no indication given as to where these measurements came from. Please provide the data in the SI or cite the paper(s) where these data came from (or, ideally, both)

We have added a table in the supplementary information (Supplementary note 4), which contains a complete list of conductivity and the corresponding references with detailed information about measurements.

A few general/high-level comments and questions:

-- It appears to me that the work is missing a compelling argument that computational predictions of the thermodynamic phase stability are actually correlated with experimental stability. Many metastable materials are synthesizable and stable at ambient conditions. This is a fairly big claim and it is central to the author's work; can the authors reference past studies that have demonstrated convincingly that this is a safe assumption?

Thank you. This specific issue has been investigated in [Science advances 2, e1600225 (2016)]. That work investigated the relation between observed stability (i.e. phases that were experimentally reported in ICSD) and E_{hull} . That work shows that while a majority of observed phases are indeed a ground state, a substantial fraction of observed compounds is not. However, these phases mostly appear in a range of 100 meV/atom above the hull (though the result is sensitive to chemistry). In addition to relying on this large-scale analysis, we also apply ideal mixing entropy to estimate the accessibility via entropy stabilization at high temperature. Therefore, we believe that the value of $E_{\text{hull}}(0\text{K}) - T S_{\text{ideal}}$ can be a useful estimator for synthetic accessibility, which is also supported by the 5/6 success rate in our experimental attempts.

However, we agree that synthetic accessibility is a complex issue that currently has no formal theory in materials science. So, given the available data and analysis, E_{hull} is the best one can do. We have added a few sentences in the result section to clarify the reliability and limitation of our current models (Page 7 and Page 12 colored in red).

-- It is stated in the introduction that synthesizability is a non-local property, and yet the authors attempt to fit local descriptors to predict it. Can the authors address this apparent contradiction?

The synthesizability is a non-local property as it involves not only the cohesive energy within the crystal structure but also the competing stability with other phases, as explained in the manuscript. Therefore, a model fitted to formation energy does not really measure the synthesizability of a material [npj Computational Materials, 2020, 6, 97]. It should be noted that in our work, we machine-learn the E_{hull} value which contains the non-local aspects of the energy as it describes the competition with phases at other composition. The physical properties to which we fit are not really local as many of them are elemental, so they have no or weak compositional dependence. Our results seem to indicate that this is enough to capture the E_{hull} behavior.

The analysis in Fig. 4 can also serve as an example to illustrate this idea. The ICOHP analysis in Fig. 4 and the discussion (page 21-page 22 colored in red) indicate that a too electronegative metal (such as Ge) leads to a lack of stability of the NASICON as the decomposition into competing phase can create a more stable bonding environment for Ge. Hence, this is a powerful example of how simple descriptors indicate that a NASICON composition would be unstable with respect to decomposition in phases with different composition (a non-local event in a phase diagram).

We have also clarified this on page 3 highlighted in red.

-- The “Physical trends of stability in NASICON” section is very good. My takeaway is that understanding the physical system in depth makes for promising descriptors, and that good descriptors are necessary for small data models. However these descriptors are very clearly not generalizable beyond the NASICON materials, despite some implications that these descriptors could provide value in other materials systems, e.g. in the introduction: “...but also

demonstrate an efficient paradigm for discovering new materials with complicate composition and atomic structure”. I don’t think this necessarily a new paradigm that’s being described here.

We would like to clarify that we are not trying to claim that the descriptors for NASICON stability would be the same for other chemical systems. On the contrary, we anticipate that such rules can differ a lot for other chemical systems. In this paper, the new paradigm we referred to is the computational framework under which phenomenological stability models and physical intuition can be mined in a data-driven fashion. Actually, it seems likely that different descriptors would be needed for structures where the bonding topology creates a different bonding competition between the elements. But these could be extracted in the same manner as presented here.

In addition, we also want to mention that even though the influence of electronegativity and miscibility do not necessarily apply to all inorganic compounds, it is still useful for understanding the stability of other materials with two distinct cation sites. There is some prior evidence in the literature that electronegativity limits on metal mixing arises from bond covalency competition: e.g ternary nitrides by Sun et al. [Nature Materials, 2019, 18, 732–739], and alkali metal-late transition metal-halides by Bartel et al. [J. Am. Chem. Soc. 2020, 142, 11, 5135–5145].

We have modified the discussion section to highlight the above arguments (colored in red in page 21-22)

Finally, a few points on the ML section. To make a compelling case, the authors first need to argue that the computational stability calculations are correlated with experimental stability (as mentioned above). Then, they need to show that the ML model is correlated with the computational data it was trained on. I’m not sure this is thoroughly shown.

The issue as to whether Ehull is a good measure of synthetic accessibility is addressed in an earlier section of this rebuttal. Our ML model is designed to optimize the capability of ranking the predicted values of Ehull as close as possible to the computed Ehull. Therefore, the best model proposed by ML should reflect the correlation between the ML model and computed Ehull values. We have clarified this issue in our discussion section [colored in red in page 23]. Finally, we also reiterate that the proof is in the pudding so as to speak. The successful synthesis of five out of six predicted compounds is a high success rate. Even the “failed” synthesis provides clear evidence of NASICON phase formation but with impurity phases. For this reason, we cannot be sure about the composition of the NASICON phase component and we classified it as a “failure”. But under less stringent evaluation criteria one can consider this a six out of six-success rate.

-- The authors appear to quote training error when providing the model performance: “Applied to all 3881 compositions, such stability model has an overall classification accuracy of 81.8%, as well as a recall value of 96.6% and F1 score of 88.2% for capturing GS/LS-NASICONs.” Was this

training error? Validation? K-fold cross-validation? Training error should almost never be cited as a model generalization error proxy.

We appreciate the comment and have made this clearer in the manuscript. To clarify, the “81.8% accuracy, 96.6% recall value and 88.2 F1 score” are for all data (80% training data + 20% validation data). The purpose of showing performance on all data is to give a sense of how our model performs on all data.

We agree that one should not use training error as a model generalization error proxy. We performed training on 80% stratified sampling data and validation on the remaining 20% of the stratified sampling data. The validation accuracy score is 84.7% while the validation F1 score is 72.3%. We have added the validation information into the revised manuscript (Page 18 colored in red).

-- The authors examined the fit of ~4M models against 3K training points. With this many “attempts” to find the best model, one risks crossing the bounds dictated by PAC learning theory. Can the authors justify this approach within the bounds of PAC theory? Otherwise, overfitting may be an issue. Although the “test” set of 32 compounds is small, there is a hint of overfitting: 96.6% recall in training but 62.5% recall on the test set.

According to PAC theory, to reduce the risk of overfitting (measured by generalization error ϵ), the amount of training data m should follow $m \propto O\left(\frac{1}{\epsilon^2} * VC(H)\right)$, where $VC(H)$ refers to the Vapnik–Chervonenkis (VC) dimension of our SIS+MLR classification model. It can be proven that the VC dimension of our classifier solely depends on the *maximal* VC dimension of the candidate descriptors. In this work, we limited the complexity of each descriptor by (a) using only 24 features to generate each descriptor, (b) combining features with only 17 basic mathematical operators. By doing this, the maximal complexity of each descriptor is fixed, and so is $VC(H)$. Using the previous formula, the required amount of training data m to avoid overfitting only depends on the maximal complexity of descriptors constructed during SIS, but not on the number of trial models. Therefore, our method of fitting more than millions of models is safe within the bounds of PAC theory. As an additional support, several other SIS/SISSO-like algorithms similar to ours also have reasonable generalizability for different chemical systems as shown in Phys. Rev. Mater. 2, 083802 (2018), J. Phys.: Mater. 2, 024002 (2019), Sci. Adv. 2019;5: eaav0693 and Nat. Commun., 9, 4168 (2018).

We regard the low 62.5% recall rate, as compared to the 96.6% training recall rate, as a result of differences in data distribution rather than overfitting. The training data set was evenly sampled across the compositional space. However, the experimentally reported NASICON data are mainly in the Na rich regime (18 out of 32 data points have $x_{Na} \geq 3$), where the stability is relatively hard to capture. Because our model optimization was done on the entire NASICON compositional space, it can negatively impact the model performance in certain local areas. This is also supported by Fig. 5c, where the Na rich compounds have lower recall rates compared with the Na poor compounds. We believe such discrepancies should not be attributed to overfitting.

References to PAC theory:

1. <https://www.cs.cmu.edu/~mgormley/courses/10601-s17/slides/lecture28-pac.pdf>
2. https://web.eecs.umich.edu/~cscott/past_courses/eecs598w14/notes/05_vc_theory.pdf
3. <http://web.cs.iastate.edu/~honavar/pac.pdf>

-- It appears the ML model made incorrect prediction(s) for the 5 successfully synthesized compounds according to Fig 5b. Can you confirm this? This would seem to challenge whether the ML model can actually predict experimental stability. I also only see two “new” materials on this plot; shouldn’t there be 6?

First of all, there are indeed 5 new NASICONs in Fig. 5b but in the descriptor space their points overlap somewhat. We did not include $\text{Na}_3\text{HfSn}(\text{SiO}_4)_2(\text{PO}_4)$ since it has a large amount of impurity during synthesis. To make those data points more distinguishable, Fig. 5b is replotted, and a list of the calculated $P_{\text{accessible}}$ of the 32 compounds is provided in supplementary note 7.

Secondly, we want to mention that 11/12 experimentally reported NASICONs with $\text{Na}_1/\text{f.u.}$ are successfully classified. While the misclassification of the 5 successfully synthesized NASICONs arises from the relatively poor performance of the model in the Na-rich regime, as mentioned in the previous comment. They are not far from predicted stability as indicated by the high Platt Scaling probability $P_{\text{accessible}}$ listed in supplementary note 7. Therefore, we believe our model is useful for predicting experimental stability. Relative modifications are added in page 18 and 19.

-- The authors discuss how their model is validated against three criteria: “1) its ability to capture experimentally reported NASICONs (Fig. 5b); 2) its robustness in identifying GS/LS NASICONs in both Na rich and Na poor regions (Fig. 5c); 3) its robustness in identifying GS/LS-NASICONs with different metal chemistries (Fig. 5d).” However, actual performance metric values are not given for #2 or #3. Can you please clarify the model performance in these areas?

We have added the actual performance metric values for #2 and #3 on page 19 and 21 (in red). For the Na-rich and Na-poor region (#2), the accuracy of the classification is 36.7% and 92.7%, respectively. Again, this is an indication that our model has better performance for Na-poor rather than Na-rich compounds. On the other hand, for Ca/Ge NASICONs and Hf/Zr NASICONs (#3), the classification accuracy is 72.9% and 75.8% respectively.

-- It is claimed a few times that the incorrect predictions are close to the decision boundary (“Moreover, the false negative classifications still remain likely to be synthetically accessible, as the average accessible probability of the false negative predictions is 30.9% for Na_1 NASICONs and 39.0% for Na_3 NASICONs.”) However, this statement is hand-wavy without knowing the distribution of other points around the decision boundary and also begs the question: then why not move the decision boundary? I believe it is better to be forthright about the model performance without discussing how the model wasn't "too wrong."

We are optimizing the F1 score in our model, so the current boundary is optimal in terms of the F1 score. Moving the boundary would increase the rate of false positives, which decreases the precision score and F1 score. In the end, where one would want this boundary is a tradeoff. In the F1 score recall and precision are treated as equally valuable, but in a realistic materials discovery program, the value of each of these would be a tradeoff between the cost of doing experiments (pushing towards higher precision) and the cost of missing out on a valuable compound (pushing towards higher recall). We have added discussion about this issue in the manuscript on Page 18, 19, 21 with edits colored in red. Additionally, the distribution of all $P_{\text{accessible}}$ is added in supplementary note 8.

Reviewer #3 (Remarks to the Author):

The manuscript by Bin Ouyang et al. present high-throughput computations for the stability and synthetic accessibility of new NASICON materials which have complex bonding topology and composition, followed with machine learning to develop a simple descriptor to describe the stability of NASICONs. Evaluation of stability and synthetic accessibility were made with the quantity of energy above the convex hull (E_{hull}) taking configurational entropy into account, and reliability of the computed data were demonstrated by experimental confirmation on six predicted new materials. This work provides significant results in that not only a new stability map of NASICONs was developed from the high-throughput computational results which is of potential importance for practical applications, but also the physical trends of the NASICONs stability was understood with the concept of bond compatibility and site miscibility and based on which a simple descriptor (“tolerance factor”) for NASICONs was identified. While this work brings novelty in the discovery and understanding of the complicated systems NASICONs, additional evidence is needed for the machine learning of the tolerance factor before I can recommend publication. My comments and suggestions are provided below.

Major comments:

1. Convergence of the machine learning results is unclear. In this work, the SIS+MLR machine learning scheme were used. The “SIS” part involves the generation of increasing complex features by using the SISO code, and the complexity was simply set to two. Actually, the feature complexity (defined as the number of operators in a feature) is a hyper-parameter that need to be optimized. It is not clear whether there is a significant improve if the feature complexity is slightly increased up to 3 or 4. Similarly, it is not discussed why the SIS-selected features were set to 2000. What will be the results if this number is increased to 2500 or 3000? Or is it because the computational cost of the MLR part limit this number to only 2000?

We thank the reviewer for bringing up those concerns. First of all, the SISO code limits feature complexity to be no more than 3 as higher complexity will lead to an overwhelmingly large combinatorial space of possible features (Phys. Rev. Materials 2, 083802). The SIS-selected subspace is also practically limited to 2000 because of memory issues.

We have tested the performance of different feature complexities and SIS-selected subspaces. Increasing complexity to 3 only improved the F1 score by 1% which led us to prefer the simpler

complexity of 2. As for SIS-selected subspace, we found that the final feature already appears among the top when the SIS-selected subspace size is set as 500. Increasing the subspace further to 2000 did not change the result of the final feature. Therefore, we think a subspace size beyond 2000 is less likely to lead to a better solution.

We have added the relevant information on page 26 colored in red.

2. In the first dimension of the 2D descriptor $t_1 = \sqrt[3]{(N_{Na})} + X_A^{Std}$, the linear combination of Na content and standard deviation of electronegativity of A is physically meaningless, though the classification in Fig 5 looks good. N_{Na} is the number of Na atoms in the chemical formula $NxM_2(AO_4)_3$, whereas the electronegativity X represents the tendency of an atom to attract electrons. It is difficult to understand why the linear combination of such two quantities is reasonable. I believe this is an important question the readers will ask if this descriptor is going to be impactful.

We do not really agree with this assessment by the reviewer. We note that the all the physical properties have been made dimensionless. Therefore, the summation of different physical factors (or their transformations) can be interpreted as an accumulation of their impact on stability. With this consideration, $t_1 = \sqrt[3]{N_{Na}} + (Q_A^{Std})^2$ (modified instead of $t_1 = \sqrt[3]{(N_{Na})} + X_A^{Std}$ as detailed in additional modification 1) indicates that Na content and polyanion charge variation penalizes the stability in this feature dimension. Though the ML procedure does not provide physical explanations, one can attempt to read meaning into these descriptors. For example, the cube root of Na content is a measure of the average Na-Na distance. The square of the charge difference between the polyanions is a measure of their average electrostatic interaction. Such an interpretation may be useful though we stress that it is not required to have a functioning model.

3. The prediction performance of the tolerance factor on unseen data is not clear. Fig 5 shows analysis of the descriptor with the training data, but I don't see a test on unseen data or cross validation (e.g. how sensitive is the descriptor on the training dataset?)

We have performed 80-20 test-validation data splitting while 20% of the data are generated from stratified sampling. Those 20% data are not used in the training process. We have highlighted our statements on page 24 colored in red. Meanwhile, we have added the specific performance of the model in both training and test data set in page 18, 19, 21 with edits colored in red.

Minor comments:

1. The authors claim the descriptor is "extremely good" in the abstract. This seems overrated as the overall training classification accuracy is 81.8%, where there is still much room to improve.

We agree that “extremely good” is overstated. We have removed “extremely” in the revised manuscript.

2. More details are needed for the MLR methods and the primary features. For example, the definition and data source for the input features ionic radii (Shannon radii?) and electronegativity (Pauling electronegativity?) is unclear. In line 279, “MLR would optimize the ranking of relative stability, which is more important than the absolute E_{hull} values ...”, please elaborate on the “relative stability” and why it is better than the absolute E_{hull} values.

Thank you for pointing out the absent of data resource. And yes, the radii are Shannon radii while the electronegativity is Pauling electronegativity. We have modified the methodology section accordingly (colored red in supplementary note 6).

The relative stability we mentioned is the order of the E_{hull} values. We trained the MLR model to maximize performance for ranking the E_{hull} values rather than to predict absolute E_{hull} values, as the relative values are more meaningful to compare the synthetic accessibility. We have also added relevant clarifications in the revised manuscript (colored in red in page 17).

Additional modifications:

1. We would like to mention that the first dimension of our machine learned descriptor is mislabeled. It should be $\sqrt[3]{N_{Na}} + (Q_A^{Std})^2$ instead of $\sqrt[3]{N_{Na}} + X_A^{Std}$. Since it was only an incorrect labelling, the numerical results are not changed. We have corrected all the labels and reference to the label across the whole manuscript (all modifications colored in red).
2. We have corrected the number of theoretically predicted synthetic accessible Na_3 NASICONs and updated Fig. 3b. The old heatmap in Fig. 3b does not include the correct estimation of ideal mixing entropy.
3. We noticed that the SVC learned boundary is inconsistent across Fig. 5, as the boundary in Fig. 5a are different from Fig. 5b-5c. It originates from the incorrect use of an older version of a figure. Fig. 5 has been updated accordingly.
4. We have also updated Fig. S6, the labels in the old figure does not have the correct dimensionless treatment.
5. The language of the whole paper is carefully revised. All the corresponding language modifications are colored in blue.

REVIEWER COMMENTS

Reviewer #1 (Remarks to the Author):

The revised version is ready for acceptance.

Reviewer #2 (Remarks to the Author):

Referee's note: I have made comments directly against the authors' most recent response document. I have not changed any of the original text; I have only added my own (new) comments, which are all preceded by ">>" characters.

The authors compute the formation energy of 3000+ NASICON materials with DFT, compute the phase diagrams to extract the energy above the convex hull, successfully synthesize 5 new materials, and then develop a two-descriptor model to describe their data.

Overall, I find the computational and experimental aspects of this work to be exciting and well-executed. The analysis of the structural factors influencing stability is similarly well done. However, I feel like the ML modeling side of the work is a bit muddled and introduces many questions. I think the work may be suitable for publication but would benefit from some clarifications and revisions before publication.

First, a few minor points:

-- Without meaning any insult to the authors, some of the English should be improved for readability, particularly in the abstract and introduction.

We thank for the reviewer for pointing out the language issue. We have carefully gone through the manuscript and polished the language. We hope such efforts had improved the readability of our manuscript.

>> The readability is much improved with these edits – looks great.

-- Krishna Rajan et al.'s 2011 work on predicting stability in perovskites should probably be cited along with the current references 12-17 (<https://doi.org/10.1098/rspa.2010.0543>)

We have added this work into our references (page 3 colored in red, literature 18)

>> Great.

-- How can you have an $E_{\text{hull}} < 0$, as referenced in Table 1?

See our answer to question 1 of Reviewer 1.

>> Thanks for providing the clarification in the text – this addresses my question.

-- The conductivities of over 300 NASICON materials are referenced in Fig. 3, but there is no indication given as to where these measurements came from. Please provide the data in the SI or cite the paper(s) where these data came from (or, ideally, both)

We have added a table in the supplementary information (Supplementary note 4), which contains a

complete list of conductivity and the corresponding references with detailed information about measurements.

>> Thanks for doing this. Providing this data all in one place will increase the impact of the paper. In the spirit of FAIR data principles (findable, accessible, interoperable and particularly reusability), the authors may also want to consider also providing the structure files in an online resources (e.g. GitHub) where they can be readily accessed by readers who wish to reproduce or improve on this work.

>> On a related note, can the authors make the rest of the data in this paper available via an online repository? E.g. the 3000+ DFT computed energies, the ML descriptors, etc.?

A few general/high-level comments and questions:

-- It appears to me that the work is missing a compelling argument that computational predictions of the thermodynamic phase stability are actually correlated with experimental stability. Many metastable materials are synthesizable and stable at ambient conditions. This is a fairly big claim and it is central to the author's work; can the authors reference past studies that have demonstrated convincingly that this is a safe assumption?

Thank you. This specific issue has been investigated in [Science advances 2, e1600225 (2016)]. That work investigated the relation between observed stability (i.e. phases that were experimentally reported in ICSD) and Ehull. That work shows that while a majority of observed phases are indeed a ground state, a substantial fraction of observed compounds is not. However, these phases mostly appear in a range of 100 meV/atom above the hull (though the result is sensitive to chemistry). In addition to relying on this large-scale analysis, we also apply ideal mixing entropy to estimate the accessibility via entropy stabilization at high temperature. Therefore, we believe that the value of Ehull(0K)-T Sideal can be a useful estimator for synthetic accessibility, which is also supported by the 5/6 success rate in our experimental attempts.

However, we agree that synthetic accessibility is a complex issue that currently has no formal theory in materials science. So, given the available data and analysis, Ehull is the best one can do. We have added a few sentences in the result section to clarify the reliability and limitation of our current models (Page 7 and Page 12 colored in red).

>> Perfect.

-- It is stated in the introduction that synthesizability is a non-local property, and yet the authors attempt to fit local descriptors to predict it. Can the authors address this apparent contradiction?

The synthesizability is a non-local property as it involves not only the cohesive energy within the crystal structure but also the competing stability with other phases, as explained in the manuscript. Therefore, a model fitted to formation energy does not really measure the synthesizability of a material [npj Computational Materials, 2020, 6, 97]. It should be noted that in our work, we machine-learn the Ehull value which contains the non-local aspects of the energy as it describes the competition with phases at other composition. The physical properties to which we fit are not really local as many of them are elemental, so they have no or weak compositional dependence. Our results seem to indicate that this is enough to capture the Ehull behavior.

The analysis in Fig. 4 can also serve as an example to illustrate this idea. The ICOHP analysis in Fig. 4 and the discussion (page 21-page 22 colored in red) indicate that a too electronegative metal (such as Ge) leads to a lack of stability of the NASICON as the decomposition into competing phase can create a more stable bonding environment for Ge. Hence, this is a powerful example of how simple

descriptors indicate that a NASICON composition would be unstable with respect to decomposition in phases with different composition (a non-local event in a phase diagram).

We have also clarified this on page 3 highlighted in red.

>> I understand the author's point here, but I don't think they have fully addressed the question. (Disclaimer: This is not a major point, and I don't wish to belabor it here, but I think it will increase the impact of the paper to address this point more clearly in the paper. Please feel free to ignore this particular comment below.)

>> I have assumed the use of the term "locality" here means that you only need to know information (structure/composition) on the phase in question, and you don't need to know anything else about the other competing phases. I believe the authors are arguing above that one can use local descriptors only to learn or infer about non-local behavior because the non-local behavior is built into the model (hyper)parameters, e.g. if you know Ge is in the structure, you know the competing phases are likely to be significantly more stable. For the record, I completely agree with the authors' point here.

>> I believe what should be said in the text is something along the lines of "synthesizability is hard to predict because it is a fundamentally nonlocal property, but it can nonetheless be learned using only local descriptors." Provided the authors do agree with this statement, I would recommend (but of course not insist) they add something to this effect in the text to clarify what appears to be a contradiction of their own making.

-- The "Physical trends of stability in NASICON" section is very good. My takeaway is that understanding the physical system in depth makes for promising descriptors, and that good descriptors are necessary for small data models. However these descriptors are very clearly not generalizable beyond the NASICON materials, despite some implications that these descriptors could provide value in other materials systems, e.g. in the introduction: "...but also demonstrate an efficient paradigm for discovering new materials with complicate composition and atomic structure". I don't think this necessarily a new paradigm that's being described here.

We would like to clarify that we are not trying to claim that the descriptors for NASICON stability would be the same for other chemical systems. On the contrary, we anticipate that such rules can differ a lot for other chemical systems. In this paper, the new paradigm we referred to is the computational framework under which phenomenological stability models and physical intuition can be mined in a data-driven fashion. Actually, it seems likely that different descriptors would be needed for structures where the bonding topology creates a different bonding competition between the elements. But these could be extracted in the same manner as presented here.

In addition, we also want to mention that even though the influence of electronegativity and miscibility do not necessarily apply to all inorganic compounds, it is still useful for understanding the stability of other materials with two distinct cation sites. There is some prior evidence in the literature that electronegativity limits on metal mixing arises from bond covalency competition: e.g ternary nitrides by Sun et al. [Nature Materials, 2019, 18, 732-739], and alkali metal-late transition metal-halides by Bartel et al. [J. Am. Chem. Soc. 2020, 142, 11, 5135-5145].

We have modified the discussion section to highlight the above arguments (colored in red in page 21-22)

>> Good edits here.

>> Referee's note: Everything in the paper up to this point is high quality work and suitable for publication in Nature Communications. However, I have major concerns when it comes to the ML work, which begins here. I am concerned not just by the quality of the work, but also by the authors' approaches and responses; going back through this paper for a second time has strengthened these concerns. I have sought to highlight my concerns below.

Finally, a few points on the ML section. To make a compelling case, the authors first need to argue that the computational stability calculations are correlated with experimental stability (as mentioned above). Then, they need to show that the ML model is correlated with the computational data it was trained on. I'm not sure this is thoroughly shown.

The issue as to whether Ehull is a good measure of synthetic accessibility is addressed in an earlier section of this rebuttal. Our ML model is designed to optimize the capability of ranking the predicted values of Ehull as close as possible to the computed Ehull. Therefore, the best model proposed by ML should reflect the correlation between the ML model and computed Ehull values. We have clarified this issue in our discussion section [colored in red in page 23]. Finally, we also reiterate that the proof is in the pudding so as to speak. The successful synthesis of five out of six predicted compounds is a high success rate. Even the "failed" synthesis provides clear evidence of NASICON phase formation but with impurity phases. For this reason, we cannot be sure about the composition of the NASICON phase component and we classified it as a "failure". But under less stringent evaluation criteria one can consider this a six out of six-success rate.

>> Upon reading the paper, I initially got the suspicion that the authors wanted to highlight the two facts that (1) DFT got 5 or 6/6 synthesizability predictions right, and (2) the ML model is good, while seeking to hide the inconvenient fact that (3) the ML model incorrectly predicted the synthesizability of all five successfully synthesized phases. The answer provided above reinforces that suspicion. "The best model proposed by ML should reflect the correlation between the ML model and the computed E_hull values... the successful synthesis of five out of six predicted compounds is a high success rate." Let's be clear: this is a high success rate for the DFT/E_hull method, but a 0% success rate for the ML model.

-- The authors appear to quote training error when providing the model performance: "Applied to all 3881 compositions, such stability model has an overall classification accuracy of 81.8%, as well as a recall value of 96.6% and F1 score of 88.2% for capturing GS/LS-NASICONs." Was this training error? Validation? K-fold cross-validation? Training error should almost never be cited as a model generalization error proxy.

We appreciate the comment and have made this clearer in the manuscript. To clarify, the "81.8% accuracy, 96.6% recall value and 88.2 F1 score" are for all data (80% training data + 20% validation data). The purpose of showing performance on all data is to give a sense of how our model performs on all data.

We agree that one should not use training error as a model generalization error proxy. We performed training on 80% stratified sampling data and validation on the remaining 20% of the stratified sampling data. The validation accuracy score is 84.7% while the validation F1 score is 72.3%. We have added the validation information into the revised manuscript (Page 18 colored in red).

>> I find this comment concerning as well. I believe this shows a fundamental misunderstanding of how ML models should be judged. The statement "the purpose of showing performance on all data is to give a sense of how our model performs on all data" does not make sense – it doesn't matter whether the performance is judged on "all data" or not, it matters whether the authors are sampling the generalization (validation or hold-out) error or not. I asked five of my colleagues working in

materials ML what they would think if they read a paper where model performance was judged by the error against an 80% training and 20% validation split, and they unanimously said this was poor practice at best and disingenuous at worst.

>> The authors have not addressed these initial concerns with the latest edits; the manuscript still cites model accuracy/recall/F1 score under 80% training. To be clear, only the performance against the 20% validation matters. Even better would be to compute a k-fold cross-validation across the entire training set.

>> I highly recommend the authors remove the statement “Applied to all 3881 compositions, this model has an overall classification accuracy of 312 81.8%, as well as a recall value of 96.6% and F1 score of 88.2% for capturing GS/LS313 NASICONs”, or they modify it to clarify the training error (only) and the validation error (only). This is a key differentiation because it enables the reader to see if there is a large divergence between the training and validation/generalization errors. If so, this would imply overfitting in the model. If not, it gives confidence that the model is not overfit. Otherwise it is difficult to contextualize the model performance.

>> As an additional metric for contextualization of model performance, can the authors cite the fraction of synthesizable/non-synthesizable in the training set? In general, a model with no statistical significance can still achieve an accuracy equal to the majority class weight – e.g. if the training set is e.g. 70% synthesizable/30% non-synthesizable, a model with zero generalizability can still validate with 70% accuracy (simply by guessing that everything is synthesizable).

-- The authors examined the fit of ~4M models against 3K training points. With this many “attempts” to find the best model, one risks crossing the bounds dictated by PAC learning theory. Can the authors justify this approach within the bounds of PAC theory? Otherwise, overfitting may be an issue. Although the “test” set of 32 compounds is small, there is a hint of overfitting: 96.6% recall in training but 62.5% recall on the test set.

According to PAC theory, to reduce the risk of overfitting (measured by generalization error ϵ), the amount of training data m should follow $m \propto O(1/\epsilon^2 * VC(H))$, where $VC(H)$ refers to the Vapnik–Chervonenkis (VC) dimension of our SIS+MLR classification model. It can be proven that the VC dimension of our classifier solely depends on the maximal VC dimension of the candidate descriptors. In this work, we limited the complexity of each descriptor by (a) using only 24 features to generate each descriptor, (b) combining features with only 17 basic mathematical operators. By doing this, the maximal complexity of each descriptor is fixed, and so is $VC(H)$. Using the previous formula, the required amount of training data m to avoid overfitting only depends on the maximal complexity of descriptors constructed during SIS, but not on the number of trial models. Therefore, our method of fitting more than millions of models is safe within the bounds of PAC theory. As an additional support, several other SIS/SISSO-like algorithms similar to ours also have reasonable generalizability for different chemical systems as shown in Phys. Rev. Mater. 2, 083802 (2018), J. Phys.: Mater. 2, 024002 (2019), Sci. Adv. 2019;5: eaav0693 and Nat. Commun., 9, 4168 (2018).

We regard the low 62.5% recall rate, as compared to the 96.6% training recall rate, as a result of differences in data distribution rather than overfitting. The training data set was evenly sampled across the compositional space. However, the experimentally reported NASICON data are mainly in the Na rich regime (18 out of 32 data points have $x_{Na} \geq 3$), where the stability is relatively hard to capture. Because our model optimization was done on the entire NASICON compositional space, it can negatively impact the model performance in certain local areas. This is also supported by Fig. 5c, where the Na rich compounds have lower recall rates compared with the Na poor compounds. We believe such discrepancies should not be attributed to overfitting.

References to PAC theory:

<https://www.cs.cmu.edu/~mgormley/courses/10601-s17/slides/lecture28-pac.pdf>

https://web.eecs.umich.edu/~cscott/past_courses/eecs598w14/notes/05_vc_theory.pdf

<http://web.cs.iastate.edu/~honaavar/pac.pdf>

>> These are good comments; I'd recommend that the authors add something to this effect to the text rather than just in this document.

>> The 62.5% test recall may very well be due to differences in the data distribution. The way to explore this is to compute the k-fold cross-validation error on the training set, which ensures you stay in the training distribution. I'd recommend the authors compute this value and cite it in the manuscript.

-- It appears the ML model made incorrect prediction(s) for the 5 successfully synthesized compounds according to Fig 5b. Can you confirm this? This would seem to challenge whether the ML model can actually predict experimental stability. I also only see two "new" materials on this plot; shouldn't there be 6?

First of all, there are indeed 5 new NASICONs in Fig. 5b but in the descriptor space their points overlap somewhat. We did not include $\text{Na}_3\text{HfSn}(\text{SiO}_4)_2(\text{PO}_4)$ since it has a large amount of impurity during synthesis. To make those data points more distinguishable, Fig. 5b is replotted, and a list of the calculated Paccessible of the 32 compounds is provided in supplementary note 7.

>> This new plot looks good.

Secondly, we want to mention that 11/12 experimentally reported NASICONs with Na1/f.u. are successfully classified. While the misclassification of the 5 successfully synthesized NASICONs arises from the relatively poor performance of the model in the Na-rich regime, as mentioned in the previous comment. They are not far from predicted stability as indicated by the high Platt Scaling probability Paccessible listed in supplementary note 7. Therefore, we believe our model is useful for predicting experimental stability. Relative modifications are added in page 18 and 19.

>> First of all, I am glad the authors' edits to page 18 now do recognize the fact that the ML model does not accurately predict the 5 newly synthesized compounds.

>> However, I am still unsettled by the apparent attempt by the authors to initially hide this fact from the reader until it was pointed out by a referee. To me, this calls into question the credibility of the authors and the work as a whole.

-- The authors discuss how their model is validated against three criteria: "1) its ability to capture experimentally reported NASICONs (Fig. 5b); 2) its robustness in identifying GS/LS NASICONs in both Na rich and Na poor regions (Fig. 5c); 3) its robustness in identifying GS/LS-NASICONs with different metal chemistries (Fig. 5d)." However, actual performance metric values are not given for #2 or #3. Can you please clarify the model performance in these areas?

We have added the actual performance metric values for #2 and #3 on page 19 and 21 (in red). For the Na-rich and Na-poor region (#2), the accuracy of the classification is 36.7% and 92.7%, respectively. Again, this is an indication that our model has better performance for Na-poor rather than Na-rich compounds. On the other hand, for Ca/Ge NASICONs and Hf/Zr NASICONs (#3), the classification accuracy is 72.9% and 75.8% respectively.

>> Thank you for adding these scores, but can the authors please clarify whether these are training errors (hopefully not) or validation errors?

-- It is claimed a few times that the incorrect predictions are close to the decision boundary ("Moreover, the false negative classifications still remain likely to be synthetically accessible, as the average accessible probability of the false negative predictions is 30.9% for Na1 NASICONs and 39.0% for Na3 NASICONs.") However, this statement is hand-wavy without knowing the distribution of other points around the decision boundary and also begs the question: then why not move the decision boundary? I believe it is better to be forthright about the model performance without discussing how the model wasn't "too wrong."

We are optimizing the F1 score in our model, so the current boundary is optimal in terms of the F1 score. Moving the boundary would increase the rate of false positives, which decreases the precision score and F1 score. In the end, where one would want this boundary is a tradeoff. In the F1 score recall and precision are treated as equally valuable, but in a realistic materials discovery program, the value of each of these would be a tradeoff between the cost of doing experiments (pushing towards higher precision) and the cost of missing out on a valuable compound (pushing towards higher recall). We have added discussion about this issue in the manuscript on Page 18, 19, 21 with edits colored in red. Additionally, the distribution of all Paccessible is added in supplementary note 8.

>> Thank you for adding this distribution to the SI; this is helpful and addresses my concerns.

Reviewer #3 (Remarks to the Author):

All of my concerns have been well addressed.

Reviewer #1 (Remarks to the Author):

The revised version is ready for acceptance.

Reviewer #2 (Remarks to the Author):

We thank the reviewer for their evaluation of our work as the revised manuscript is much improved as a result. Before we address each point individually, we want to summarize our responses to the reviewer's major concerns:

1. **Transparency of the data:** We do not agree with the reviewer that we are trying to hide any data (details below in specific comment). We have made all raw data publicly available and have done so in the revised manuscript via a GitHub repo together with Jupyter notebooks to reproduce our ML model (details below).
2. **Validation method:** With respect to the validation method, we agree that reporting scores that combine training and validation data is not a good practice. We have now done 5-fold cross validation that shows an average validation accuracy of 82.1%, which is close to the validation accuracy of 82.4% with the 80-20 splitting data set used in model selection (SIS+MLR). We agree with the reviewer that K-fold cross validation is a more rigorous way to evaluate our model, and details about the newly performed 5-fold cross validation have been added into the revised manuscript.
3. **Rigorousness of the modeling process:** We regret the disorganization of the machine learning part. There were indeed parts with unclear information. We have now performed a rigorous check of all the contents in this section, specifically addressing the points raised by the reviewer and also identifying a few additional points of clarification highlighted in the end of the response letter.

With the above-mentioned modifications, we are confident that the machine learning part of the paper is greatly improved, and all of our arguments are both rigorous and reproducible. For the following one-to-one response, our response is highlighted with red color.

Referee's note: I have made comments directly against the authors' most recent response document. I have not changed any of the original text; I have only added my own (new) comments, which are all preceded by ">>" characters.

The authors compute the formation energy of 3000+ NASICON materials with DFT, compute the phase diagrams to extract the energy above the convex hull, successfully synthesize 5 new materials, and then develop a two-descriptor model to describe their data.

Overall, I find the computational and experimental aspects of this work to be exciting and well-executed. The analysis of the structural factors influencing stability is similarly well done. However, I feel like the ML modeling side of the work is a bit muddled and introduces many

questions. I think the work may be suitable for publication but would benefit from some clarifications and revisions before publication.

First, a few minor points:

-- Without meaning any insult to the authors, some of the English should be improved for readability, particularly in the abstract and introduction.

We thank for the reviewer for pointing out the language issue. We have carefully gone through the manuscript and polished the language. We hope such efforts had improved the readability of our manuscript.

>> The readability is much improved with these edits – looks great.

-- Krishna Rajan et al.'s 2011 work on predicting stability in perovskites should probably be cited along with the current references 12-17 (<https://doi.org/10.1098/rspa.2010.0543>)

We have added this work into our references (page 3 colored in red, literature 18)

>> Great.

-- How can you have an $E_{\text{hull}} < 0$, as referenced in Table 1?

See our answer to question 1 of Reviewer 1.

>> Thanks for providing the clarification in the text – this addresses my question.

-- The conductivities of over 300 NASICON materials are referenced in Fig. 3, but there is no indication given as to where these measurements came from. Please provide the data in the SI or cite the paper(s) where these data came from (or, ideally, both)

We have added a table in the supplementary information (Supplementary note 4), which contains a complete list of conductivity and the corresponding references with detailed information about measurements.

>> Thanks for doing this. Providing this data all in one place will increase the impact of the paper. In the spirit of FAIR data principles (findable, accessible, interoperable and particularly reusability), the authors may also want to consider also providing the structure files in an online resources (e.g. GitHub) where they can be readily accessed by readers who wish to reproduce or improve on this work.

>> On a related note, can the authors make the rest of the data in this paper available via an online repository? E.g. the 3000+ DFT computed energies, the ML descriptors, etc.?

Yes. All the DFT computed energies and ML descriptors have been put online as a Github repo https://github.com/Jeff-oakley/NASICON_Predictor_Data. In addition to that, we have also uploaded Jupyter notebooks that can be used to reproduce both the 80-20 validation and 5-fold cross validation procedures that we refer to in the revised manuscript. A detailed description is also provided in the Data availability section of the revised manuscript on Page 28 highlighted with red color.

A few general/high-level comments and questions:

-- It appears to me that the work is missing a compelling argument that computational predictions of the thermodynamic phase stability are actually correlated with experimental stability. Many metastable materials are synthesizable and stable at ambient conditions. This is a fairly big claim and it is central to the author's work; can the authors reference past studies that have demonstrated convincingly that this is a safe assumption?

Thank you. This specific issue has been investigated in [Science advances 2, e1600225 (2016)]. That work investigated the relation between observed stability (i.e. phases that were experimentally reported in ICSD) and Ehull. That work shows that while a majority of observed phases are indeed a ground state, a substantial fraction of observed compounds is not. However, these phases mostly appear in a range of 100 meV/atom above the hull (though the result is sensitive to chemistry). In addition to relying on this large-scale analysis, we also apply ideal mixing entropy to estimate the accessibility via entropy stabilization at high temperature. Therefore, we believe that the value of Ehull(OK)-T Sideal can be a useful estimator for synthetic accessibility, which is also supported by the 5/6 success rate in our experimental attempts.

However, we agree that synthetic accessibility is a complex issue that currently has no formal theory in materials science. So, given the available data and analysis, Ehull is the best one can do. We have added a few sentences in the result section to clarify the reliability and limitation of our current models (Page 7 and Page 12 colored in red).

>> Perfect.

-- It is stated in the introduction that synthesizability is a non-local property, and yet the authors attempt to fit local descriptors to predict it. Can the authors address this apparent contradiction?

The synthesizability is a non-local property as it involves not only the cohesive energy within the crystal structure but also the competing stability with other phases, as explained in the manuscript. Therefore, a model fitted to formation energy does not really measure the synthesizability of a material [npj Computational Materials, 2020, 6, 97]. It should be noted that in our work, we machine-learn the Ehull value which contains the non-local aspects of the energy as it describes the competition with phases at other composition. The physical properties to which we fit are not really local as many of them are elemental, so they have no or weak compositional dependence. Our results seem to indicate that this is enough to capture the Ehull behavior.

The analysis in Fig. 4 can also serve as an example to illustrate this idea. The ICOHP analysis in Fig. 4 and the discussion (page 21-page 22 colored in red) indicate that a too electronegative metal (such as Ge) leads to a lack of stability of the NASICON as the decomposition into competing phase can create a more stable bonding environment for Ge. Hence, this is a powerful example of how simple descriptors indicate that a NASICON composition would be unstable with respect to decomposition in phases with different composition (a non-local event in a phase diagram).

We have also clarified this on page 3 highlighted in red.

>> I understand the author's point here, but I don't think they have fully addressed the question. (Disclaimer: This is not a major point, and I don't wish to belabor it here, but I think it will increase the impact of the paper to address this point more clearly in the paper. Please feel free to ignore this particular comment below.)

>> I have assumed the use of the term "locality" here means that you only need to know information (structure/composition) on the phase in question, and you don't need to know anything else about the other competing phases. I believe the authors are arguing above that one can use local descriptors only to learn or infer about non-local behavior because the non-local behavior is built into the model (hyper)parameters, e.g. if you know Ge is in the structure, you know the competing phases are likely to be significantly more stable. For the record, I completely agree with the authors' point here.

>> I believe what should be said in the text is something along the lines of "synthesizability is hard to predict because it is a fundamentally nonlocal property, but it can nonetheless be learned using only local descriptors." Provided the authors do agree with this statement, I would recommend (but of course not insist) they add something to this effect in the text to clarify what appears to be a contradiction of their own making.

We agree that this point should be clarified. We have added the following statement to page 3 of the revised manuscript: "Synthesizability is hard to predict because it is a fundamentally nonlocal property [Bartel et. al., *npj Computational Materials* 5, 4, (2019)], but it can nonetheless be learned using only local descriptors, specifically when the scope of materials under investigation is restricted to a single structure or class of structures [Bartel et. al., *Science advances* 5, eaav0693 (2019)]."

-- The "Physical trends of stability in NASICON" section is very good. My takeaway is that understanding the physical system in depth makes for promising descriptors, and that good descriptors are necessary for small data models. However these descriptors are very clearly not generalizable beyond the NASICON materials, despite some implications that these descriptors could provide value in other materials systems, e.g. in the introduction: "...but also demonstrate an efficient paradigm for discovering new materials with complicate composition and atomic structure". I don't think this necessarily a new paradigm that's being described here.

We would like to clarify that we are not trying to claim that the descriptors for NASICON stability would be the same for other chemical systems. On the contrary, we anticipate that such rules can differ a lot for other chemical systems. In this paper, the new paradigm we referred to is the computational framework under which phenomenological stability models and physical intuition can be mined in a data-driven fashion. Actually, it seems likely that different descriptors would be needed for structures where the bonding topology creates a different bonding competition between the elements. But these could be extracted in the same manner as presented here.

In addition, we also want to mention that even though the influence of electronegativity and miscibility do not necessarily apply to all inorganic compounds, it is still useful for understanding the stability of other materials with two distinct cation sites. There is some prior evidence in the literature that electronegativity limits on metal mixing arises from bond covalency competition: e.g ternary nitrides by Sun et al. [Nature Materials, 2019, 18, 732–739], and alkali metal-late transition metal-halides by Bartel et al. [J. Am. Chem. Soc. 2020, 142, 11, 5135–5145].

We have modified the discussion section to highlight the above arguments (colored in red in page 21-22)

>> Good edits here.

>> Referee's note: Everything in the paper up to this point is high quality work and suitable for publication in Nature Communications. However, I have major concerns when it comes to the ML work, which begins here. I am concerned not just by the quality of the work, but also by the authors' approaches and responses; going back through this paper for a second time has strengthened these concerns. I have sought to highlight my concerns below.

Finally, a few points on the ML section. To make a compelling case, the authors first need to argue that the computational stability calculations are correlated with experimental stability (as mentioned above). Then, they need to show that the ML model is correlated with the computational data it was trained on. I'm not sure this is thoroughly shown.

The issue as to whether Ehull is a good measure of synthetic accessibility is addressed in an earlier section of this rebuttal. Our ML model is designed to optimize the capability of ranking the predicted values of Ehull as close as possible to the computed Ehull. Therefore, the best model proposed by ML should reflect the correlation between the ML model and computed Ehull values. We have clarified this issue in our discussion section [colored in red in page 23]. Finally, we also reiterate that the proof is in the pudding so as to speak. The successful synthesis of five out of six predicted compounds is a high success rate. Even the "failed" synthesis provides clear evidence of NASICON phase formation but with impurity phases. For this reason, we cannot be sure about the composition of the NASICON phase component and we classified it as a "failure". But under less stringent evaluation criteria one can consider this a six out of six-success rate.

>> Upon reading the paper, I initially got the suspicion that the authors wanted to highlight the two facts that (1) DFT got 5 or 6/6 synthesizability predictions right, and (2) the ML model is good,

while seeking to hide the inconvenient fact that (3) the ML model incorrectly predicted the synthesizability of all five successfully synthesized phases. The answer provided above reinforces that suspicion. “The best model proposed by ML should reflect the correlation between the ML model and the computed E_{hull} values... the successful synthesis of five out of six predicted compounds is a high success rate.” Let's be clear: this is a high success rate for the DFT/ E_{hull} method, but a 0% success rate for the ML model.

We apologize for the lack of clarity in our previous response; however, we do not agree with the reviewer that we are intentionally seeking to hide the fact that the ML model misclassified the 5 NASICONs newly synthesized in this work. In our previous submission, this point has already been stated clearly:

1. On Page 18 of the manuscript, “... we applied the [ML] model to classify 27 previously reported and 5 newly synthesized NASICONs (shown in Fig 5b)... The model correctly captures 20 of the 32 experimentally synthesized NASICONs as synthetically accessible... **the five newly synthesized Na_3 -NASICONs are all misclassified when the boundary is set at $P_{\text{accessible}} = 50\%$.**” This final sentence states exactly what the reviewer purports we are trying to hide.
2. In Figure 5b, the five inverted green triangles are the data points the reviewer purports are being hidden. These points are further emphasized with an inset in this figure.

Nonetheless, we have further expanded this discussion in the revised manuscript to avoid readers having the same interpretation. While more statements concerning this point can be found at Page 18–19 (colored in red). Our main modifications are summarized below:

1. We strengthened the point in the revised abstract and the main text (colored red on Page 12) that 5/6 successful rate for newly synthesized NASICON is solely for E_{hull} capability of capturing synthetic accessibility.
2. The fact that 0/5 rate of success in predicting newly synthesis compound is highlighted and discussed both in Fig. 5 and page 19. However, we interpret the 0/5 rate as weak performance for capturing Na rich compounds rather than overfitting as 1) 5-fold cross validation suggests no overfitting; 2) The detailed comparison between Na rich and Na poor performance in page 19-20 and Fig. 5c shows obvious performance difference among Na rich and Na poor compounds. Additionally, there is no sign of overfitting even compare the recall of the LS/GS NASICONs in the training set with the validation set within Na poor or Na rich region (Page 20).

-- The authors appear to quote training error when providing the model performance: “Applied to all 3881 compositions, such stability model has an overall classification accuracy of 81.8%, as well as a recall value of 96.6% and F1 score of 88.2% for capturing GS/LS-NASICONs.” Was this training error? Validation? K-fold cross-validation? Training error should almost never be cited as a model generalization error proxy.

We appreciate the comment and have made this clearer in the manuscript. To clarify, the “81.8% accuracy, 96.6% recall value and 88.2 F1 score” are for all data (80% training data + 20% validation data). The purpose of showing performance on all data is to give a sense of how our model performs on all data.

We agree that one should not use training error as a model generalization error proxy. We performed training on 80% stratified sampling data and validation on the remaining 20% of the stratified sampling data. The validation accuracy score is 84.7% while the validation F1 score is 72.3%. We have added the validation information into the revised manuscript (Page 18 colored in red).

>> I find this comment concerning as well. I believe this shows a fundamental misunderstanding of how ML models should be judged. The statement “the purpose of showing performance on all data is to give a sense of how our model performs on all data” does not make sense – it doesn’t matter whether the performance is judged on “all data” or not, it matters whether the authors are sampling the generalization (validation or hold-out) error or not. I asked five of my colleagues working in materials ML what they would think if they read a paper where model performance was judged by the error against an 80% training and 20% validation split, and they unanimously said this was poor practice at best and disingenuous at worst.

>> The authors have not addressed these initial concerns with the latest edits; the manuscript still cites model accuracy/recall/F1 score under 80% training. To be clear, only the performance against the 20% validation matters. Even better would be to compute a k-fold cross-validation across the entire training set.

We agree with the reviewer such statements can be misleading. We have removed all statements about the model performance on all data. Instead, we only cite validation metrics for evaluating our model performance throughout the revised manuscript (Page 17-20 in red color).

>> I highly recommend the authors remove the statement “Applied to all 3881 compositions, this model has an overall classification accuracy of 81.8%, as well as a recall value of 96.6% and F1 score of 88.2% for capturing GS/LS313 NASICONs”, or they modify it to clarify the training error (only) and the validation error (only). This is a key differentiation because it enables the reader to see if there is a large divergence between the training and validation/generalization errors. If so, this would imply overfitting in the model. If not, it gives confidence that the model is not overfit. Otherwise it is difficult to contextualize the model performance.

Similar with our above response. We agree that we should not report the overall performance of the model on all data. In the revised manuscript, we followed the reviewer’s suggestion to use the 5-fold cross validation on determining our decision boundary, and only the metrics for the validation set is used. Meanwhile, when evaluating the performance of our final model, we compared also metrics on training data with validation data to check if there is any obvious divergence. As being shown in page 18 and supplementary note 7, the training accuracy (82.3%) and F1 score (74.9%) are quite similar to validation accuracy (84.5%) and F1 score (78.1%). This further confirms no overfitting.

>> As an additional metric for contextualization of model performance, can the authors cite the fraction of synthesizable/non-synthesizable in the training set? In general, a model with no statistical significance can still achieve an accuracy equal to the majority class weight – e.g. if the training set is e.g. 70% synthesizable/30% non-synthesizable, a model with zero generalizability can still validate with 70% accuracy (simply by guessing that everything is synthesizable).

In our data, ~83% of the computation data is unstable (US), so it is true that a naïve model that only predicts US would achieve an accuracy of 83%. However, this same model would achieve a recall of 0% (and an F1 score of 0). In our model, we achieve 81.3% recall for predicting GS/LS NASICONS, demonstrating a significant improvement over this naïve baseline. However, we agree that one should be careful when evaluating performance with unbalanced data. To address this, we 1) used class-weighted SVC and stratified sampling to overcome the uneven amount of GS/LS and US NASICONS (colored in red in methods section on page 28); 2) When evaluating performance, we use accuracy, recall and f1 score to give a more systematic evaluate model performance.

We agree that comparing to a naïve baseline and discussing the unbalanced data is important, so we have cited the fraction of synthesizable/non-synthesizable in the data set in the revised manuscript (colored in red in methods) and discussed our interpretation of the validation accuracy in methods and supplementary note 7.

-- The authors examined the fit of ~4M models against 3K training points. With this many “attempts” to find the best model, one risks crossing the bounds dictated by PAC learning theory. Can the authors justify this approach within the bounds of PAC theory? Otherwise, overfitting may be an issue. Although the “test” set of 32 compounds is small, there is a hint of overfitting: 96.6% recall in training but 62.5% recall on the test set.

According to PAC theory, to reduce the risk of overfitting (measured by generalization error ϵ), the amount of training data m should follow $m \propto O(1/\epsilon^2 * VC_{\text{total}}(H))$, where $VC(H)$ refers to the Vapnik–Chervonenkis (VC) dimension of our SIS+MLR classification model. It can be proven that the VC dimension of our classifier solely depends on the maximal VC dimension of the candidate descriptors. In this work, we limited the complexity of each descriptor by (a) using only 24 features to generate each descriptor, (b) combining features with only 17 basic mathematical operators. By doing this, the maximal complexity of each descriptor is fixed, and so is $VC_{\text{total}}(H)$. Using the previous formula, the required amount of training data m to avoid overfitting only depends on the maximal complexity of descriptors constructed during SIS, but not on the number of trial models. Therefore, our method of fitting more than millions of models is safe within the bounds of PAC theory. As an additional support, several other SIS/SISSO-like algorithms similar to ours also have reasonable generalizability for different chemical systems as shown in Phys. Rev. Mater. 2, 083802 (2018), J. Phys.: Mater. 2, 024002 (2019), Sci. Adv. 2019;5: eaav0693 and Nat. Commun., 9, 4168 (2018).

We regard the low 62.5% recall rate, as compared to the 96.6% training recall rate, as a result of differences in data distribution rather than overfitting. The training data set was evenly sampled

across the compositional space. However, the experimentally reported NASICON data are mainly in the Na rich regime (18 out of 32 data points have $x_{\text{Na}} \geq 3$), where the stability is relatively hard to capture. Because our model optimization was done on the entire NASICON compositional space, it can negatively impact the model performance in certain local areas. This is also supported by Fig. 5c, where the Na rich compounds have lower recall rates compared with the Na poor compounds. We believe such discrepancies should not be attributed to overfitting.

References to PAC theory:

<https://www.cs.cmu.edu/~mgormley/courses/10601-s17/slides/lecture28-pac.pdf>

https://web.eecs.umich.edu/~cscott/past_courses/eecs598w14/notes/05_vc_theory.pdf

<http://web.cs.iastate.edu/~honavar/pac.pdf>

>> These are good comments; I'd recommend that the authors add something to this effect to the text rather than just in this document.

Yes. We have added them in the revised manuscript as well (Page 27).

>> The 62.5% test recall may very well be due to differences in the data distribution. The way to explore this is to compute the k-fold cross-validation error on the training set, which ensures you stay in the training distribution. I'd recommend the authors compute this value and cite it in the manuscript.

We have done 5-fold cross validation in the revised manuscript (see our response to comments above). Meanwhile, we have also evaluated the data at different Na contents and metal chemistry to explore the performance for different data distributions. We found the Na content can lead to the largest variation on accuracy. That also explains the 62.5% recall (56.3% in the updated model from 5-fold CV) in capturing 32 experimentally reported compounds as 18 of them are Na rich NASICONs (for which the model was found to perform relatively poorly). Detailed discussions are added in revised manuscript at Page 18-20.

-- It appears the ML model made incorrect prediction(s) for the 5 successfully synthesized compounds according to Fig 5b. Can you confirm this? This would seem to challenge whether the ML model can actually predict experimental stability. I also only see two "new" materials on this plot; shouldn't there be 6?

First of all, there are indeed 5 new NASICONs in Fig. 5b but in the descriptor space their points overlap somewhat. We did not include $\text{Na}_3\text{HfSn}(\text{SiO}_4)_2(\text{PO}_4)$ since it has a large amount of impurity during synthesis. To make those data points more distinguishable, Fig. 5b is replotted, and a list of the calculated Paccessible of the 32 compounds is provided in supplementary note 7.

>> This new plot looks good.

Secondly, we want to mention that 11/12 experimentally reported NASICONs with $\text{Na}_1/\text{f.u.}$ are successfully classified. While the misclassification of the 5 successfully synthesized NASICONs

arises from the relatively poor performance of the model in the Na-rich regime, as mentioned in the previous comment. They are not far from predicted stability as indicated by the high Platt Scaling probability Pacesible listed in supplementary note 7. Therefore, we believe our model is useful for predicting experimental stability. Relative modifications are added in page 18 and 19.

>> First of all, I am glad the authors' edits to page 18 now do recognize the fact that the ML model does not accurately predict the 5 newly synthesized compounds.

>> However, I am still unsettled by the apparent attempt by the authors to initially hide this fact from the reader until it was pointed out by a referee. To me, this calls into question the credibility of the authors and the work as a whole.

Same with our response above. Starting from the initial draft, the fact that the five newly synthesized compounds are not predicted successfully has already been shown in Fig. 5b and listed in supplementary note 5. We also made further modifications in our first revision by zooming in all five points to show specifically those five compounds are not classified correctly by the model. In this second revision, we have made new edits to ensure this point is clear (colored red in 18).

-- The authors discuss how their model is validated against three criteria: "1) its ability to capture experimentally reported NASICONs (Fig. 5b); 2) its robustness in identifying GS/LS NASICONs in both Na rich and Na poor regions (Fig. 5c); 3) its robustness in identifying GS/LS-NASICONs with different metal chemistries (Fig. 5d)." However, actual performance metric values are not given for #2 or #3. Can you please clarify the model performance in these areas?

We have added the actual performance metric values for #2 and #3 on page 19 and 21 (in red). For the Na-rich and Na-poor region (#2), the accuracy of the classification is 36.7% and 92.7%, respectively. Again, this is an indication that our model has better performance for Na-poor rather than Na-rich compounds. On the other hand, for Ca/Ge NASICONs and Hf/Zr NASICONs (#3), the classification accuracy is 72.9% and 75.8% respectively.

>> Thank you for adding these scores, but can the authors please clarify whether these are training errors (hopefully not) or validation errors?

Those were scores for all data (training + testing). As mentioned in the above comments, we have now removed all statements of performance on all data and only cited validation metrics for evaluating performance.

-- It is claimed a few times that the incorrect predictions are close to the decision boundary ("Moreover, the false negative classifications still remain likely to be synthetically accessible, as the average accessible probability of the false negative predictions is 30.9% for Na1 NASICONs and 39.0% for Na3 NASICONs.") However, this statement is hand-wavy without knowing the distribution of other points around the decision boundary and also begs the question: then why

not move the decision boundary? I believe it is better to be forthright about the model performance without discussing how the model wasn't "too wrong."

We are optimizing the F1 score in our model, so the current boundary is optimal in terms of the F1 score. Moving the boundary would increase the rate of false positives, which decreases the precision score and F1 score. In the end, where one would want this boundary is a tradeoff. In the F1 score recall and precision are treated as equally valuable, but in a realistic materials discovery program, the value of each of these would be a tradeoff between the cost of doing experiments (pushing towards higher precision) and the cost of missing out on a valuable compound (pushing towards higher recall). We have added discussion about this issue in the manuscript on Page 18, 19, 21 with edits colored in red. Additionally, the distribution of all Paccessible is added in supplementary note 8.

>> Thank you for adding this distribution to the SI; this is helpful and addresses my concerns.

Additional clarifications and corrections of the ML model:

1. In the first revision, we have the statement that "The validation accuracy score is 84.7% while the validation F1 score is 72.3%." (Page 17) We noticed that the reviewer may think this refers to the validation error of the SVC model that determines the classification boundary. We want to clarify that this is the validation error of the machine learning ranking model for evaluating ranking performance of the 2D descriptor. Such metrics can be reproduced in Run_Ranked_SVM.ipynb
2. We agree with the reviewer's suggestion that 5-fold cross validation will be a great way to evaluate the ML model. We have used that for determining the decision boundary between LS/GS and US NASICONs. However, we still keep the use of 80-20 splitting for selecting 2D descriptors. Our reasons are 1) The 80-20 splitting is a widely used way for evaluating a model and our comparison between 80-20 splitting and 5-fold cross validation for the fitting of the class-weighted SVC gives quite similar validation accuracy (shown in our first response and Run_five_fold_CV.ipynb); 2) Repeating the 2D descriptor selection on the 2 million feature space during k-fold CV is quite computationally expensive (many orders of magnitude more than fitting the SVC once the 2D descriptor is fixed).

Reviewer #3 (Remarks to the Author):

All of my concerns have been well addressed.

REVIEWERS' COMMENTS

Reviewer #4 (Remarks to the Author):

The manuscript by Ouyang et al. discusses stability rules for NASICON-structured materials. This work combines many pieces of data from high-throughput DFT calculations, machine learning, and experiment.

My evaluation of this manuscript is focused on the machine learning (ML) section of the paper. I agree with reviewer #2 that the ML part is not very strong and had severe issues concerning the validation of the models. The ML model also does not predict the newly discovered materials to be stable, so that it does not add much to the manuscript in addition to DFT. For the revision, the authors performed the 5-fold cross validation suggested by reviewer #2 and now distinguish clearly between model performance on the training and test sets. This is very good and makes the ML validation more believable.

However, the paragraphs added to the manuscript are not well written, they suffer from poor English, lack logical structure, and make use of technical jargon, which makes them hard to understand. In order to make sense of it all, I also took a look at the GitHub repository with the provided data. Unfortunately, the data is simply dumped and not at all documented, so that it is not particularly useful. Overall, I therefore have to conclude that the ML part of this work is not yet up to the standards of Nature Communications. I have no questions in addition to those of reviewer #2, but the authors should carefully rewrite the ML section of the manuscript, moving the technical details to the SI while keeping the concepts in the main paper. The shared data on GitHub should additionally be documented such that the work and especially the model validation become fully reproducible.

Response to the Reviewers: Manuscript ID NCOMMS-21-06620B

We thank the editor and fourth reviewer for their time spent to evaluate our manuscript and for their recommendation/comments. The reviewer's concerns are addressed below in detail. All changes made in the manuscript are highlighted in red.

Reviewer #4 (Remarks to the Author):

The manuscript by Ouyang et al. discusses stability rules for NASICON-structured materials. This work combines many pieces of data from high-throughput DFT calculations, machine learning, and experiment.

My evaluation of this manuscript is focused on the machine learning (ML) section of the paper. I agree with reviewer #2 that the ML part is not very strong and had severe issues concerning the validation of the models. The ML model also does not predict the newly discovered materials to be stable, so that it does not add much to the manuscript in addition to DFT. For the revision, the authors performed the 5-fold cross validation suggested by reviewer #2 and now distinguish clearly between model performance on the training and test sets. This is very good and makes the ML validation more believable.

However, the paragraphs added to the manuscript are not well written, they suffer from poor English, lack logical structure, and make use of technical jargon, which makes them hard to understand. In order to make sense of it all, I also took a look at the GitHub repository with the provided data. Unfortunately, the data is simply dumped and not at all documented, so that it is not particularly useful. Overall, I therefore have to conclude that the ML part of this work is not yet up to the standards of Nature Communications. I have no questions in addition to those of reviewer #2, but the authors should carefully rewrite the ML section of the manuscript, moving the technical details to the SI while keeping the concepts in the main paper. The shared data on GitHub should additionally be documented such that the work and especially the model validation become fully reproducible.

We would like to thank the reviewer for the careful evaluation and good suggestions. We have gone through the manuscript, particularly machine learning section thoroughly. As suggested by the reviewer, the logical structure and English writing were carefully revised. Some technical details were moved to the SI as well. We believe the readability of the manuscript is much improved. The Github repo (https://github.com/Jeff-oakley/NASICON_Predictor_Data) has been carefully organized as well. All the data and demonstration scripts are documented so people can reproduce our data more efficiently.